# The role of *Limch1* alternative splicing in skeletal muscle function

Matthew S Penna[1,4] , Rong-Chi Hu[2] , George G Rodney[2] , Thomas A Cooper[1,2,3]

**Postnatal skeletal muscle development is a highly dynamic period associated with widespread alternative splicing changes required to adapt tissues to adult function. These splicing events have significant implications because the reversion of adult mRNA isoforms to fetal isoforms is observed in forms of muscular dystrophy. LIMCH1 is a stress fiber–associated protein that is alternatively spliced to generate uLIMCH1, a ubiquitously expressed isoform, and mLIMCH1, a skeletal muscle–specific isoform containing six additional exons simultaneously included after birth in the mouse. CRISPR/Cas9 was used to delete the six alternatively spliced exons of LIMCH1 in mice, thereby forcing the constitutive expression of the predominantly fetal isoform, uLIMCH1. mLIMCH1 knockout mice had significant grip strength weakness in vivo, and maximum force generated was decreased ex vivo. Calcium-handling deficits were observed during myofiber stimulation that could explain the mechanism by which mLIMCH1 knockout leads to muscle weakness. In addition, *LIMCH1* is mis-spliced in myotonic dystrophy type 1, with the muscleblind-like (MBNL) family of proteins acting as the likely major regulator of *Limch1* alternative splicing in skeletal muscle.**

## Introduction

Postnatal skeletal muscle development is a period of dynamic change as the tissue undergoes extensive remodeling to meet new functional demands. A significant aspect of this process is the regulated alternative splicing of hundreds of genes, many of which are temporally coordinated. This process facilitates the transition from fetal to adult protein isoforms integral in the development of adult tissue functionality (Giudice et al, 2014; Brinegar et al, 2017; Mazin et al, 2021). Although large numbers of alternative splicing transitions have been identified in genome-wide transcriptomic analyses, the functional contribution of individual isoform transitions to muscle physiology remains largely unexplored (Buckingham et al, 2003; Aulehla & Pourquié, 2008; Braun &

Gautel, 2011; Kalsotra & Cooper, 2011). To identify alternative splicing transitions with likely functional importance, we used criteria including conservation of the alternative exons, temporal conservation of splicing pattern transitions, developmental regulation, and coding potential (Keren et al, 2010; Barbosa-Morais et al, 2012; Ellis et al, 2012). In previous studies, we demonstrated extensive alternative splicing transitions during postnatal skeletal muscle development and redirected splicing of four vesicular trafficking genes toward fetal isoforms in adult skeletal muscle, resulting in T-tubule disruption, calcium-handling deficits, and impaired force generation (Giudice et al, 2016; Brinegar et al, 2017). Mis-regulated alternative splicing also has pathogenic consequences in skeletal muscle (Pistoni et al, 2010). A notable example is the disease myotonic dystrophy type 1 (DM1), in which perturbed activities of RNA-binding proteins muscleblind-like (MBNL) and CUG-BP, Elav-like family (CELF) leads to a reversion to fetal alternative splicing patterns for hundreds of genes, resulting in subsequent muscle wasting (Mankodi et al, 2002; Turner & Hilton-Jones, 2010; Wang et al, 2019).

LIM and calponin homology domains 1 (*LIMCH1*) is a ubiquitously expressed gene with mRNA and protein expression in skeletal, cardiac, brain, respiratory, gastrointestinal, and reproductive tissues (Uhlén et al, 2015; Sjöstedt et al, 2020). The LIMCH1 protein contains one calponin homology (CH) domain at the N-terminus, several coiled-coil domains located throughout the middle of the protein, and one LIM domain located at the C-terminal region, which canonically has roles in actin binding, protein oligomerization, and protein–protein interactions, respectively (Kadrmas & Beckerle, 2004; Park, 2020; Yin et al, 2020). The sole study investigating the function of LIMCH1 characterized it as an actin stress fiber–associated protein that binds non-muscle myosin 2A (NM2A) to regulate focal adhesion formation. Silencing of LIMCH1 led to disrupted focal adhesions and a subsequent increase in cell migration (Lin et al, 2017). Several reports have linked LIMCH1 depletion to cancer progression, suggesting reduced LIMCH1 levels may serve as a biomarker for tumor growth and metastasis in lung, breast, and cervical cancers (Cizkova et al, 2010; Karlsson et al, 2018; Zhang et al, 2019; Bersini et al, 2020; Cao et al, 2021; Halle et al, 2021). In the context of skeletal muscle, a study examining myofibers

[1]Department of Pathology & Immunology, Baylor College of Medicine, Houston, TX, USA   [2]Department of Integrative Physiology, Baylor College of Medicine, Houston, TX, USA   [3]Department of Molecular & Cellular Biology, Baylor College of Medicine, Houston, TX, USA   [4]Medical Scientist Training Program, Baylor College of Medicine, Houston, TX, USA

Correspondence: tcooper@bcm.edu; rodney@bcm.edu

affected by neurogenic muscular atrophy, identified the over-expression of 55 proteins, with LIMCH1 being one of them and found most of them were involved in myofibrillogenesis (Krause et al, 2022). Beyond the role of LIMCH1 in focal adhesion formation and cell migration in vitro, nothing is known about its role in tissue-specific development or physiology.

In adult mouse skeletal muscle, *Limch1* is alternatively spliced, leading to the coordinated and skeletal muscle–specific inclusion of six exons not found in the ubiquitous isoform of LIMCH1, uLIMCH1. This muscle-specific LIMCH1 isoform, mLIMCH1, is the predominant isoform in adult skeletal muscle. The six exons introduce 454 in-frame amino acids predicted to be disordered with no identifiable domains, which is a common feature of alternatively spliced protein regions (Romero et al, 2006). The functional advantage of coordinately including six exons in LIMCH1 to form a novel, skeletal muscle–specific, adult-predominant protein isoform is unknown. We hypothesized that mLIMCH1 contributes to adult muscle function and homeostasis based on three key features: developmental regulation involving a fetal to adult isoform transition, highly specific expression of the mLIMCH1 isoform in skeletal muscle, and evolutionary conservation of mLIMCH1 isoform across multiple mammalian species.

In this study, we used CRISPR/Cas9 to delete the genomic segment containing the six skeletal muscle–specific exons of *Limch1* in mice, leading to the constitutive expression of uLIMCH1 throughout postnatal development and in adult mouse skeletal muscle. In mice lacking mLIMCH1, we identified significant muscle weakness in vivo without loss of muscle mass, and significant reductions of maximum force, rate of relaxation, and rate of contraction demonstrated ex vivo. We demonstrate differential localization of uLIMCH1 and mLIMCH1 and show that LIMCH1 preferentially localizes to the sarcolemma region in WT myofibers. This preferential localization is lost in isolated myofibers in the absence of mLIMCH1 with uLIMCH1 primarily localized throughout the sarcoplasm of the myofiber. In addition, myofibers lacking mLIMCH1 exhibit disrupted calcium handling, which we propose as a mechanism contributing to muscle weakness. We also show that inclusion of the six skeletal muscle–specific exons requires MBNL activity, and LIMCH1 splicing is disrupted in patient-derived DM1 skeletal muscle tissue revealing the loss of mLIMCH1 expression as a novel mis-regulated splicing event in DM1 muscle.

## Results

### Alternative splicing generates a skeletal muscle–specific isoform of LIMCH1 during postnatal development

We analyzed our previously published transcriptomic datasets for alternative splicing changes during postnatal skeletal muscle development in mice to identify undescribed protein isoform transitions likely to be important for adult tissue function (Brinegar et al, 2017). We identified a conserved splicing transition in the *Limch1* gene that leads to the coordinated inclusion of six exons that introduces 454 in-frame internal amino acids expressed exclusively in skeletal muscle (Fig 1A). The ubiquitous isoform, uLIMCH1, is expressed in most tissues; however, in skeletal muscle,

mLIMCH1 expression increases soon after birth to become the predominant isoform in adults (Fig 1B). RT-PCR of WT mouse skeletal muscle tissue indicates a developmental switch from *uLimch1*, the predominant isoform in fetal skeletal muscle, to *mLimch1*, the predominant isoform in adult skeletal muscle (Fig 1C). In postnatal day 1 (PN1) skeletal muscle, the six exons unique to *mLimch1* are included in only ~33% of the total *Limch1* mRNA, with *uLimch1* constituting the remaining ~67% of the *Limch1* mRNA transcripts. However, in adult skeletal muscle, the *mLimch1* isoform constitutes most (~72%) of the total *Limch1* mRNA (Fig 1C and D).

The muscle-specific region in humans is encoded by seven exons constituting 544 amino acids with 65% homology to the skeletal muscle–specific region in mouse *Limch1*. RT–PCR analysis demonstrates that the developmentally regulated inclusion of the muscle-specific region is conserved between human and mice. As in mouse, *mLIMCH1* is the predominant isoform in human adult skeletal muscle whereas *uLIMCH1* predominates in human fetal skeletal muscle (Fig 1E and F).

### CRISPR-mediated deletion of the skeletal muscle–specific exons in *Limch1*

To determine the extent to which the developmentally regulated and conserved splicing of *Limch1* contributes to skeletal muscle physiology, we used CRISPR/Cas9 to delete *Limch1* exons 9–14 that encode the *mLimch1* isoform from the genome of FVB mice, thereby leading to the expression of only *uLimch1* in adult skeletal muscle of mice homozygous (HOM) for the deleted allele (*Limch1* 6exKO) (Fig 2A). RT–PCR confirms the absence of the six alternatively spliced exons of *Limch1* and the correct splicing of the flanking constitutive exons 8 and 15 that are included in RNA from *Limch1* 6exKO skeletal muscle tissue (Fig 2B). Using an antibody that recognizes both isoforms, we performed western blotting for endogenous uLIMCH1 (~135 kD) and mLIMCH1 (~200 kD) protein levels in different skeletal muscle tissues (Fig 2C). HOM *Limch1* 6exKO mice lack expression of mLIMCH1 and exclusively express uLIMCH1. As expected from mRNA data, mLIMCH1 protein is undetectable in WT cardiac and brain tissue (Fig 2D). Two independent founder lines from the CRISPR/Cas9 deletion of *Limch1* exons 9–14 were generated and shown to have comparable phenotypes, ruling out the likelihood of CRISPR/Cas9-mediated off-target effects. In addition, the line of *Limch1* 6exKO mice used for experiments was derived from F1 animals and then backcrossed for seven generations to out-cross off-target loci. One of the lines was selected for the subsequent phenotyping assays.

### Exclusive expression of uLIMCH1 in adult skeletal muscle leads to reduced grip strength and muscle force generation

To assess the physiological consequences of the loss of mLIMCH1 in adult skeletal muscle, we evaluated the muscle function of HOM *Limch1* 6exKO mice side-by-side with WT age–matched controls. We first measured grip strength in adult HOM *Limch1* 6exKO mice. Results showed a significant decrease in the grip strength of HOM *Limch1* 6exKO mice, with a reduction of 31% and 29% in male and female mice, respectively (Fig 3A). We evaluated maximum force production in both the extensor digitorum longus (EDL) (primarily

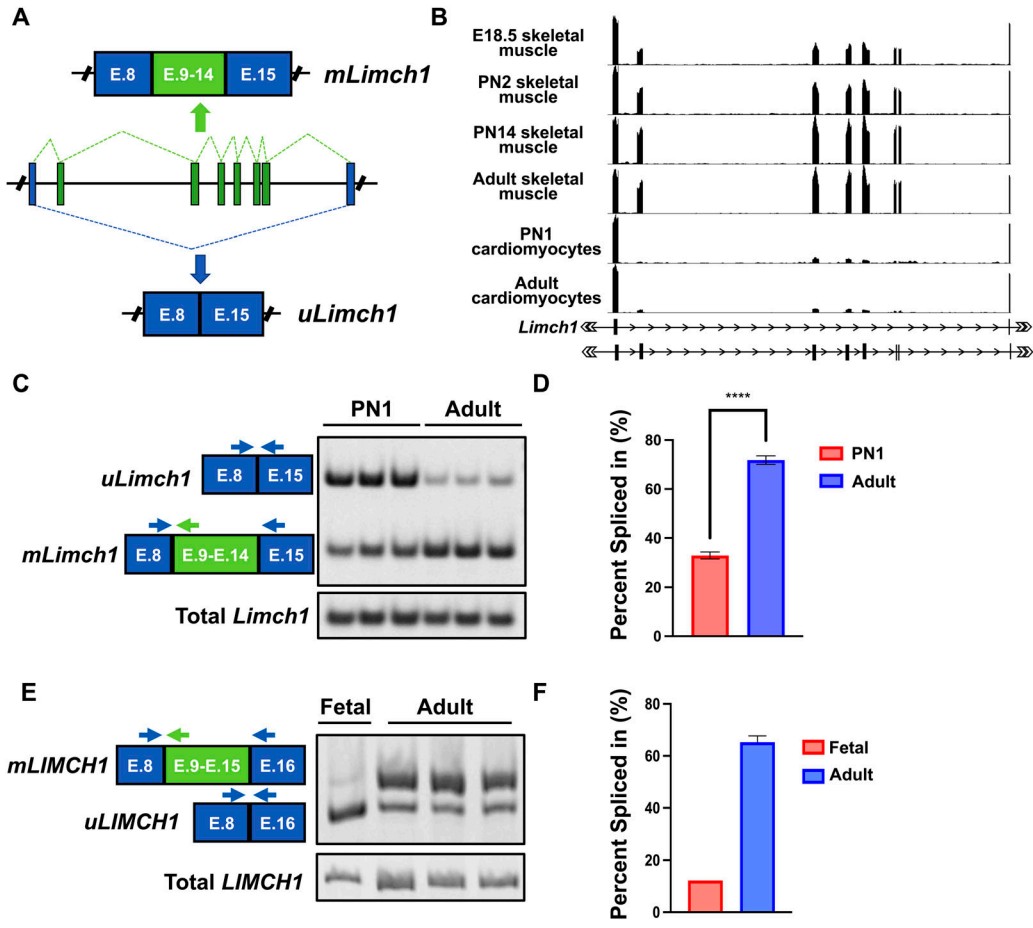

**Figure 1. Skeletal muscle–specific inclusion of six exons (E.9–E.14) in *Limch1* during postnatal development.**
**(A)** Alternative splicing of the *Limch1* pre-mRNA in skeletal muscle produces a ubiquitously expressed mRNA isoform, *uLimch1*, and a skeletal muscle–specific isoform, *mLimch1*. **(B)** RNA-seq tracks of *Limch1* mRNA from mouse skeletal and cardiac muscle at different ages: E18.5 skeletal muscle, PN2 skeletal muscle, PN14 skeletal muscle, adult skeletal muscle, PN1 cardiomyocytes, and adult cardiomyocytes (Giudice et al, 2014; Brinegar et al, 2017). **(C)** RT–PCR showing the *Limch1* alternative splicing pattern in PN1 and adult skeletal muscle tissue. **(D)** Quantification of the *Limch1* muscle–specific exon inclusion in PN1 and adult skeletal muscle (n = 3 PN1, 3 adult). **(E)** RT–PCR showing the *LIMCH1* alternative splicing pattern in fetal and adult human skeletal muscle tissue. **(F)** Quantification of the *LIMCH1* muscle–specific exon inclusion in fetal and adult human skeletal muscle (n = 1 fetal, 3 adult). Data represent the mean ± SEM and were analyzed using a two-tailed *t* test. ****P < 0.0001. E, exon; PN1, postnatal day 1; PN2, postnatal day 2; E18.5, embryonic day 18.5; PN14, postnatal day 14.

fast-twitch fibers) and soleus (predominantly slow-twitch fibers) of HOM *Limch1* 6exKO mice and found a statistically significant decrease in the maximum force produced with a particularly stronger reduction in force generation at higher frequencies (Fig 3B and C). We next conducted isometric ex vivo force analysis of contractile parameters on EDL and soleus muscles to measure multiple components of muscle function directly. The rate of relaxation and rate of contraction of a 1 Hz stimulus leading to a twitch were significantly reduced in the EDL from HOM *Limch1* 6exKO mice compared with WT age–matched controls (Fig 3D and E). The rate of contraction and relaxation in the soleus was not significantly different between HOM *Limch1* 6exKO and WT mice (Fig S1A and B). No differences in fatigue or recovery after repeated stimulation were observed in EDL or soleus tissue from HOM *Limch1* 6exKO mice (Fig S1C–H).

We conducted H&E and picrosirius red staining on muscle cross sections collected from HOM *Limch1* 6exKO tissue to determine whether underlying histological defects could be contributing to

the defects in muscle function. HOM *Limch1* 6exKO mice did not exhibit differences in myofiber diameter, centralized nuclei, or fibrotic scarring compared with WT control mice at 10–12 wk of age, and no overt morphological differences were observed in older Limch1 6exKO mice at 1 yr of age. In addition, no differences in muscle weights were observed between the HOM *Limch1* 6exKO and WT control mice (Fig S2). These results indicate that muscle weakness did not correlate with skeletal muscle histopathology or muscle loss, suggesting an intrinsic defect in muscle contraction.

### Localization of LIMCH1 is disrupted in HOM *Limch1* 6exKO myofibers

To elucidate whether loss of the muscle-specific exons impacted LIMCH1 localization, we isolated flexor digitorum brevis (FDB) myofibers from WT and HOM *Limch1* 6exKO mice and performed immunofluorescence staining to identify endogenous localization patterns of mLIMCH1 and uLIMCH1 using an antibody that

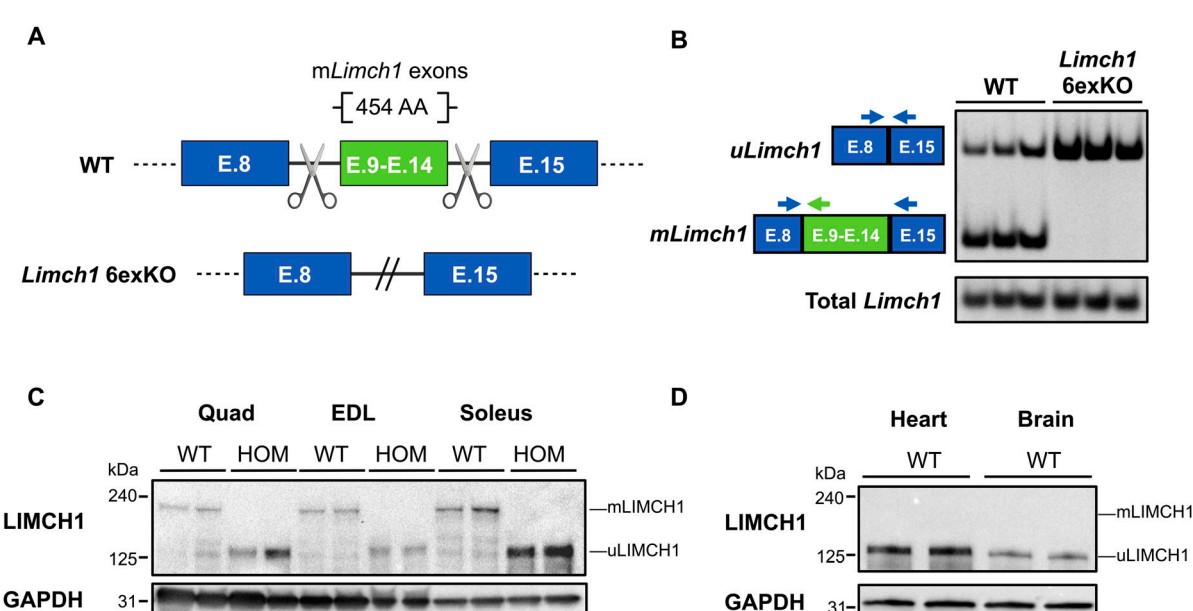

**Figure 2. CRISPR/Cas9-mediated germline deletion of exons 9–14 to force the expression of the fetal predominant isoform, uLIMCH1.**
**(A)** Diagram of the CRISPR/Cas9 approach to delete *Limch1* muscle–specific exons 9–14 from the mouse genome. Scissors represent the guide RNAs targeting the intronic regions flanking the *mLimch1*-specific exons. **(B)** RT–PCR showing *Limch1* alternative splicing pattern in WT and HOM *Limch1* 6exKO skeletal muscle highlighting the removal of the six skeletal muscle–specific exons in *Limch1* 6exKO tissue. **(C)** LIMCH1 protein isoform expression in adult HOM *Limch1* 6exKO mice and WT age–matched controls in quad, EDL, and soleus skeletal muscle (n = 2 WT, 2 HOM). **(D)** Western blot showing LIMCH1 protein expression patterns in the heart and brain of WT mice (n = 2 WT). E, exon; AA, amino acid; HOM, homozygous; WT, wild-type; quad, quadriceps; EDL, extensor digitorum longus.

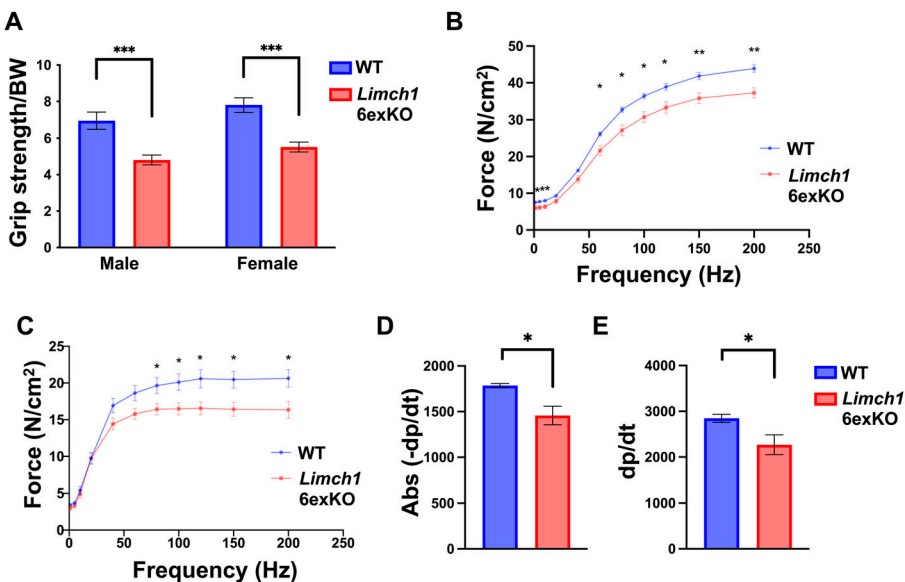

**Figure 3. Loss of the muscle-specific LIMCH1 isoform produces grip strength weakness and force production deficits in HOM *Limch1* 6exKO mice.**
**(A)** Four-limb grip strength assessment in HOM *Limch1* 6exKO mice and WT age–matched controls (n = 13 WT, 13 HOM males; 12 WT, 14 HOM females). **(B)** Force–frequency curve during isometric stimulation of the EDL stimulated at increasing frequencies. **(C)** Force–frequency curve during isometric stimulation of the soleus stimulated at increasing frequencies. **(D)** Rate of relaxation (-dp/dt) in the EDL following an ex vivo isometric twitch. **(E)** Rate of contraction (dp/dt) in the EDL following an ex vivo isometric twitch. (For (B, C, D, E), EDL: n = 6 WT, 8 HOM; soleus: n = 6 WT, 6 HOM). Data represent the mean ± SEM and were analyzed using a two-tailed *t* test. Force–frequency curves were analyzed using multiple *t* tests for each frequency tested. *$P < 0.05$, **$P < 0.01$, ***$P < 0.001$. BW, body weight; Abs, absolute value; WT, wild-type; HOM, homozygous; EDL, extensor digitorum longus.

recognizes a shared epitope. The strongest LIMCH1 signal in WT myofibers is observed at the sarcolemma. In HOM *Limch1* 6exKO myofibers, LIMCH1 staining is also observed at the sarcolemma but lacks the stark signal difference between the sarcolemma and sarcoplasm observed in WT myofibers (Fig 4A).

We quantified the immunofluorescence signal with plot profiles collected along the sarcolemma of the myofiber (along red arrows in Fig 4A), which showed that LIMCH1 is distributed in a punctate manner with maximum peak intensity occurring every two microns in both WT and HOM *Limch1* 6exKO myofibers (Fig 4B, left panels). We also conducted plot profiles to compare staining along the sarcolemma in WT and HOM *Limch1* 6exKO myofibers (blue line in Fig 4A). The signal intensity in WT myofibers was stronger along the sarcolemma compared with the sarcoplasm of the myofiber, whereas HOM *Limch1* 6exKO myofibers had a more even distribution from the sarcolemma to the sarcoplasm in the myofiber

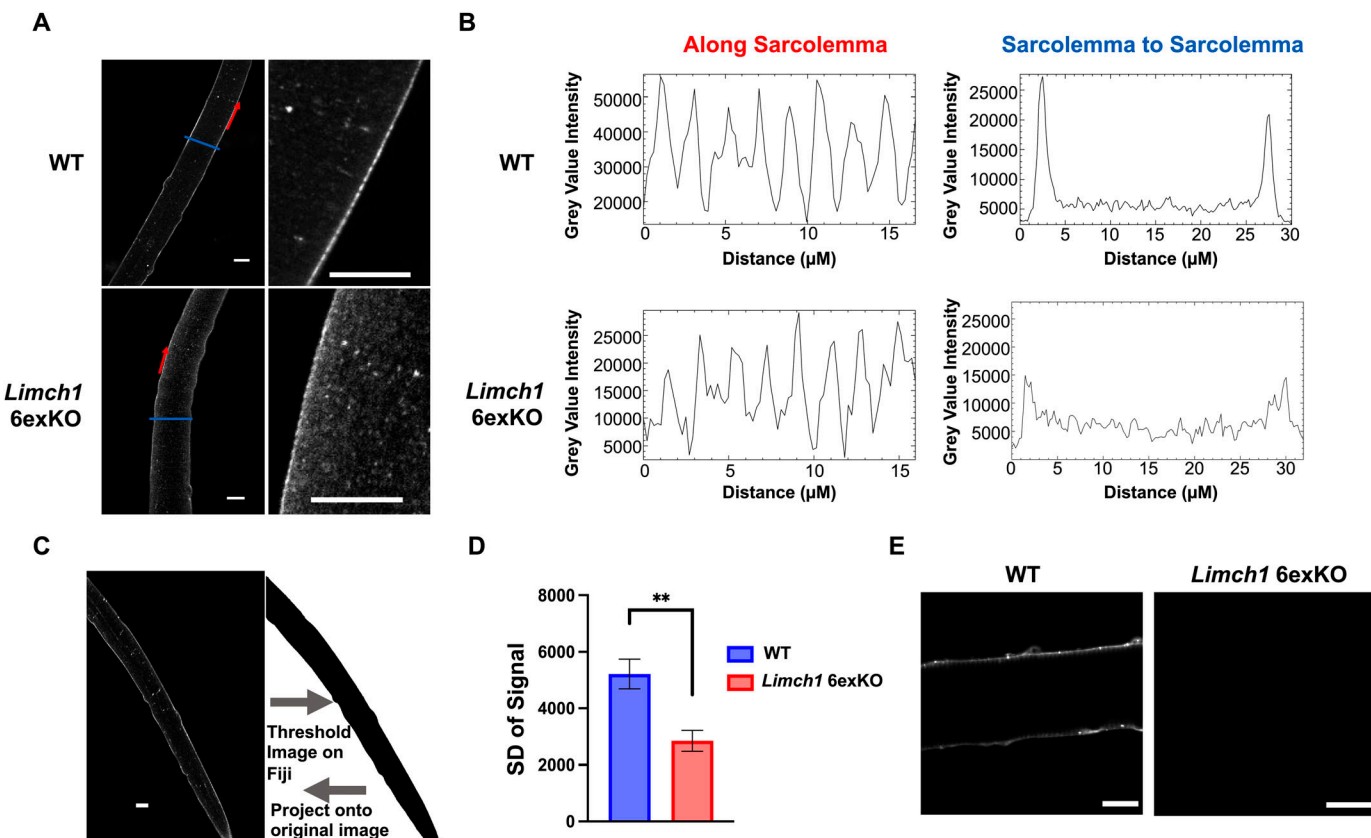

**Figure 4. LIMCH1 is differentially localized in HOM *Limch1* 6exKO myofibers.**
**(A)** Immunofluorescence staining of endogenous LIMCH1 in WT and HOM *Limch1* 6exKO FDB myofibers with zoomed-in panels on the right. Scale bar, 15 μM.
**(A, B)** Representative signal intensity plot profile along the sarcolemma of WT and HOM *Limch1* 6exKO myofibers (red arrows in (A), left panel of (B)) and from sarcolemma to sarcolemma of the myofiber in WT and HOM *Limch1* 6exKO myofibers (blue line in (A), right panel of (B)). **(C)** Approach for analyzing signal distribution using Fiji to turn the image into a binary image and project the identified signal onto the original image for analysis. Scale bar, 15 μM. **(D)** SD of overall LIMCH1 signal in WT and HOM *Limch1* 6exKO myofibers from isolated FDB myofibers (multiple myofibers averaged from n = 6 WT, 6 HOM *Limch1* 6exKO mice). **(E)** Immunofluorescence staining of endogenous mLIMCH1 in WT and HOM *Limch1* 6exKO FDB myofibers. Scale bar, 15 μM. Data represent the mean ± SEM and were analyzed using a two-tailed *t* test. **\*\*P < 0.01.** WT, wild-type; HOM, homozygous; SD, standard deviation; FDB, flexor digitorum brevis.

(Fig 4B, right panels). We analyzed LIMCH1 immunofluorescence throughout the entire myofiber to determine whether the LIMCH1 signal was differentially localized in WT and HOM *Limch1* 6exKO myofibers as the plot profile comparing sarcolemmas suggests. After determining the signal area by thresholding the image and projecting the outline back onto the original image for analysis, we calculated the SD of each pixel (signal) within the myofiber to quantitate signal distribution (Fig 4C). The SD of the signal from HOM *Limch1* 6exKO myofibers was decreased significantly compared with WT myofibers indicating that the stark differential staining in the sarcolemma and sarcoplasm of WT myofibers is consistently observed in myofibers lacking mLIMCH1, suggesting preferential staining along the sarcolemma by mLIMCH1 (Fig 4D).

To further delineate the differential localization of mLIMCH1 and uLIMCH1, we developed an mLIMCH1-specific antibody to differentiate uLIMCH1 and mLIMCH1 isoforms and better understand the precise localization of mLIMCH1 in skeletal muscle myofibers. The antibody was commercially synthesized, and we validated the antibody for Western blots and immunofluorescence detection (Fig S3). Myofibers stained with the mLIMCH1-specific antibody revealed that mLIMCH1 localization heavily favors the sarcolemma. uLIMCH1

is unable to recapitulate normal sarcolemma localization upon mLIMCH1 knockout, and the increased cytoplasmic signal observed in *Limch1* 6exKO myofibers is likely from the increased expression of uLIMCH1 as the sole isoform produced from the deletion alleles (Fig 4E). Taken together, these results demonstrate that mLIMCH1 preferentially localizes to the sarcolemma of skeletal muscle myofibers with differential localization of the fetal predominant LIMCH1 isoform in adult HOM *Limch1* 6exKO myofibers.

**Intracellular calcium handling is altered in HOM *Limch1* 6exKO skeletal muscle**

We conducted ex vivo calcium analysis during a myofiber twitch and tetanus to investigate the potential contributions of altered calcium handling to the skeletal muscle dysfunction observed in vivo and ex vivo. Calcium directly regulates muscle contraction whereas defects in calcium handling can impact force generation and lead to muscle weakness (Berchtold et al, 2000; Westerblad et al, 2010; Kuo & Ehrlich, 2015). We found that peak calcium levels were decreased in male and female HOM *Limch1* 6exKO–derived myofibers stimulated at 1 Hz (Fig 5A), 20 Hz (Fig 5B), and 100 Hz (Fig 5C)

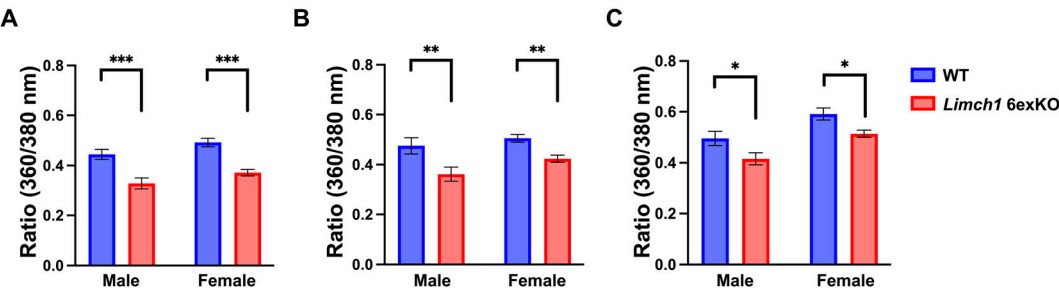

**Figure 5.   Calcium handling is disrupted in HOM *Limch1* 6exKO–isolated myofibers.**
**(A)** Peak Fura ratio (360/380 nm) in FDB myofibers stimulated with a 1 Hz twitch. **(B)** Peak Fura ratio (360/380 nm) in FDB myofibers stimulated with a 20 Hz tetanus.
**(C)** Peak Fura ratio (360/380 nm) in FDB myofibers stimulated with a 100 Hz tetanus (2–3 fibers averaged from n = 10 WT, 11 HOM *Limch1* 6exKO males; 9 WT, 9 *Limch1* 6exKO females). Data represent the mean ± SEM and were analyzed using a two-tailed *t* test. *$P < 0.05$, **$P < 0.01$, ***$P < 0.001$. nm, nanometer; WT, wild-type; FDB, flexor digitorum brevis; HOM, homozygous.

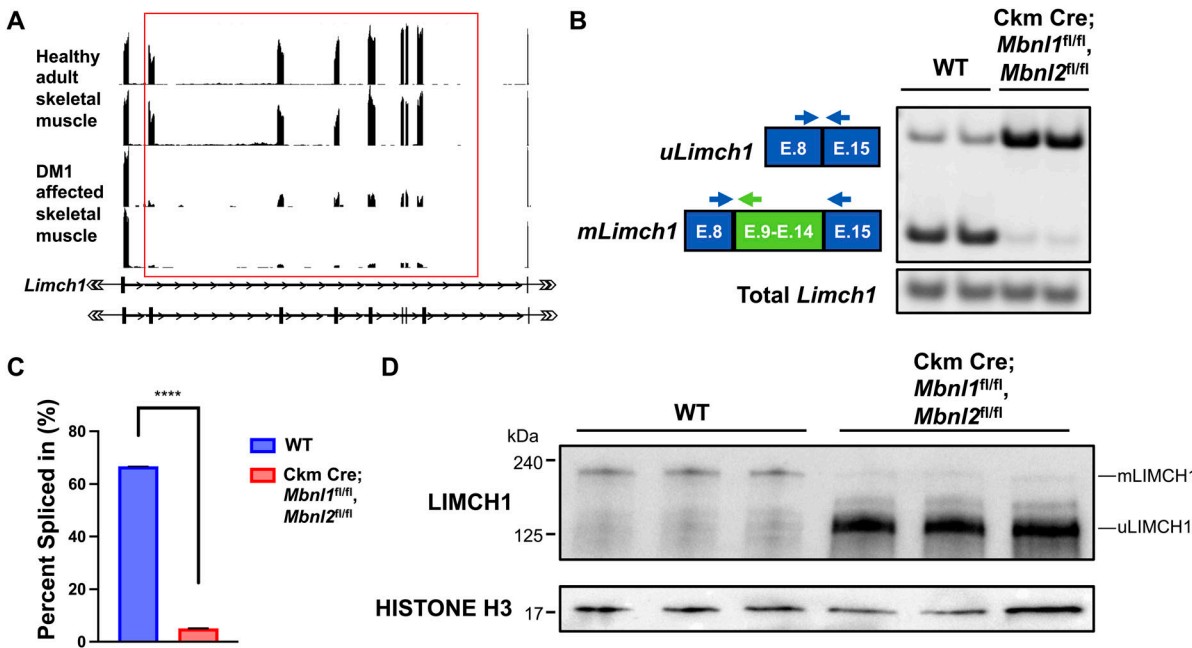

**Figure 6.   *LIMCH1* is mis-spliced in human DM1 skeletal muscle.**
**(A)** RNA-seq tracks of *LIMCH1* from DM1 and non-DM1 human skeletal muscle. The red box indicates the seven skeletal muscle–specific exons found in the human gene and are mis-spliced in DM1 skeletal muscle. **(B)** RT–PCR showing *Limch1* alternative splicing pattern in WT and Ckm Cre; *Mbnl1*^fl/fl, *Mbnl2*^fl/fl mouse skeletal muscle.
**(C)** Quantification of *Limch1* muscle–specific exon inclusion in Ckm Cre; *Mbnl1*^fl/fl, *Mbnl2*^fl/fl skeletal muscle (n = 2 WT, 2 Ckm Cre; *Mbnl1*^fl/fl, *Mbnl2*^fl/fl). **(D)** LIMCH1 protein expression in Ckm Cre; *Mbnl1*^fl/fl, *Mbnl2*^fl/fl skeletal muscle (n = 3 WT, 3 Ckm Cre; *Mbnl1*^fl/fl, *Mbnl2*^fl/fl). Data represent the mean ± SEM and were analyzed using a two-tailed *t* test. ****$P < 0.0001$. DM1, myotonic dystrophy type 1; E, exon; WT, wild-type.

frequencies compared with WT myofibers. Results showed an average peak calcium decrease of 26.1%, 23.8%, and 16.0% in male-derived myofibers at 1, 20, and 100 Hz, respectively, whereas results showed an average peak calcium decrease of 24.4%, 16.1%, and 13.2% in female derived myofibers at 1, 20, and 100 Hz, respectively. We followed up the peak calcium analysis with immunofluorescence of different T-tubule–related proteins to determine whether T-tubule structure is intact in HOM *Limch1* 6exKO myofibers and found intact T-tubule structure (Fig S4). We find that in the absence of mLIMCH1, calcium handling is disrupted in skeletal muscle whereas the gross structural integrity of the T-tubule system remains intact.

### *LIMCH1* is aberrantly spliced in DM1

Aberrant alternative splicing is a hallmark of many skeletal muscle diseases, with many of the affected genes expressing specific isoforms tailored for skeletal muscle function (Mankodi et al, 2002; Charlet-B. et al, 2002; Montes et al, 2019). We found that the *LIMCH1* pre-mRNA is mis-spliced in skeletal muscle tissue from adults affected with DM1. In humans, the skeletal muscle–specific region of *LIMCH1* constitutes seven exons encoding 544 amino acids with 65% homology with the mouse segment. DM1 skeletal muscle lacks the appropriate inclusion of these exons and expresses the fetal isoform, *uLIMCH1*, as the predominant isoform (Fig 6A). MBNL1 and

MBNL2 are RNA-binding proteins that regulate hundreds of genes through alternative splicing, and MBNL protein sequestration is directly involved in DM1 pathogenesis (Lee & Cooper, 2009; Konieczny et al, 2014). To better understand the mechanism by which *LIMCH1* alternative splicing is regulated, we looked at LIMCH1 expression in floxed (fl) Ckm Cre; *Mbnl1*fl/fl, *Mbnl2*fl/fl mice (Fig S5). Using RT–PCR, in adult Ckm Cre; *Mbnl1*fl/fl, *Mbnl2*fl/fl skeletal muscle tissue, we found that only ~5% of total *Limch1* mRNA included the six skeletal muscle–specific exons compared with ~66% in age–matched WT skeletal muscle tissue (Fig 6B and C). Consistent with mRNA expression, we found significantly decreased mLIMCH1 protein expression and a reversion to the fetal predominant protein isoform, uLIMCH1 in adult Ckm Cre; *Mbnl1*fl/fl, *Mbnl2*fl/fl skeletal muscle tissue (Fig 6D). We conclude from the results that the MBNL family plays a significant role in the tissue-specific regulation of LIMCH1, and MBNL sequestration is the probable mechanism by which *LIMCH1* is mis-spliced in DM1.

## Discussion

Alternative splicing is a mode of post-transcriptional gene regulation that contributes to the production of novel isoforms in a wide variety of tissues, thereby leading to the functional diversification of the proteome (Pan et al, 2008). Further exploration is needed to understand the role of tissue-specific alternative splicing transitions and the pathological consequences of their mis-regulation in striated muscle. By screening for evolutionarily conserved and developmentally regulated alternative splicing transitions in skeletal muscle, we identified a postnatal splicing transition in the *Limch1* gene. This unique splicing event is characterized by the coordinated inclusion of six exons constituting 454 in-frame amino acids and leads to the production of a novel skeletal muscle–specific isoform, mLIMCH1.

In this work, we sought to establish the contribution of mLIMCH1 to skeletal muscle function. We generated mice homozygous for a genomic deletion of the six exons encoding the muscle-specific isoform, thereby exclusively expressing the fetal predominant isoform, uLIMCH1, in adult skeletal muscle. These mice demonstrated significant skeletal muscle weakness in vivo and deficits in skeletal muscle force generation ex vivo. We found decreased force generation in both the slow-twitch muscle (soleus) and fast-twitch muscle (EDL).

Importantly, we saw no differences in muscle architecture based on histology and no muscle wasting based on muscle weights of the HOM *Limch1* 6exKO mice. These results provide evidence that global disruptions in muscle morphology are not contributing to the reduced force production and muscle weakness. It is likely that mLIMCH1 is contributing to normal muscle function through the maintenance of interfibrillar properties intrinsic to the muscle fiber.

To understand the specialized function of mLIMCH1, we used immunofluorescent microscopy to establish the isoform-specific endogenous localization of mLIMCH1 in the muscle myofiber. We found that LIMCH1 had preferential localization to the sarcolemma in WT myofibers with repeated staining observed at two-micron intervals. To differentiate localization between mLIMCH1 and

uLIMCH1 isoforms, we synthesized an mLIMCH1 isoform–specific antibody and found that mLIMCH1 is observed predominantly in the sarcolemma. This localization is altered in HOM *Limch1* 6exKO myofibers, where uLIMCH1 exhibited a differential localization in the sarcoplasm instead of the sarcolemma, underscoring the necessity of the skeletal muscle–specific isoform for normal LIMCH1 localization.

The disruption of calcium handling can alter skeletal muscle contraction in the absence of overt morphological changes (Berchtold et al, 2000). The decreased rate of relaxation and rate of contraction in EDL from HOM *Limch1* 6exKO mice suggests calcium handling as a mechanism of the observed muscle weakness. With the gross morphology of skeletal muscle intact in HOM *Limch1* 6exKO, we show that mLIMCH1 is functionally significant for calcium handling, with deficits in peak calcium release during myofiber stimulation likely involved in the resulting muscle weakness. T-tubule–related proteins in the HOM *Limch1* 6exKO myofibers appear structurally intact, which necessitates future studies on the mechanism by which mLIMCH1 removal affects homeostatic calcium handling.

We also identified *LIMCH1* mis-splicing in DM1 muscle and showed that loss of MBNL activity, as observed in DM1, leads to the mis-regulation of the *Limch1* muscle–specific exons. Many genes are mis-spliced in DM1, although some pathogenic splice variants are directly implicated in disease pathology, the contribution of other variants is not well understood. In this study, we did not directly examine the role *LIMCH1* plays in DM1 progression. However, future studies will determine the degree to which *LIMCH1* mis-splicing contributes to strength and force loss in DM1.

There are certain factors from our study that should be taken into consideration. In HOM *Limch1* 6exKO skeletal muscle tissue, higher uLIMCH1 expression may contribute to skeletal muscle deficits because the same amount of pre-mRNA is produced, and thus more uLIMCH1 protein will be produced than normally observed in adult tissue. Furthermore, there is an inherent challenge in studying the protein binding characteristics of mLIMCH1 and structural contributions of mLIMCH1 to muscle cell architecture due to the lack of mLIMCH1 expression in cultured myoblasts or differentiated myotubes likely attributed to the immature state of these cells even when fully differentiated. With this study establishing the functional significance of mLIMCH1 in skeletal muscle, future work is focused on identifying the specific protein interactors of mLIMCH1 and uLIMCH1 facilitating skeletal muscle function both during development and in adult tissue.

## Materials and Methods

### Animals

For all animal studies, NIH guidelines for the use and care of laboratory animals approved by the Baylor College of Medicine Institutional Animal Care and Use Committee were followed. Heterozygous *Limch1* 6exKO mice were crossed to produce litters with homozygous *Limch1* 6exKO mice and WT mice. The WT mice are offspring from the founder line that established the *Limch1* 6exKO

mouse line and were used in all experiments. All mice used in the experiments were 10–12 wk old except when otherwise noted. Ckm Cre; Mbnl1[fl/fl], Mbnl2[fl/fl] tissue used in RT–PCR and Western blot experiments were collected from a previously unpublished mouse line. In collaboration with the BCM Mouse Embryonic Stem Cell Core (mES Core), loxP sites flanking exon three in Mbnl1 and exon two in Mbnl2 were introduced, and mice were appropriately crossed to produce Mbnl1[fl/fl], Mbnl2[fl/fl] mice. These mice were subsequently crossed with a Ckm Cre line (B6.FVB(129S4)-Tg(Ckmm-cre)5Khn/J; Jackson Labs 006475) to induce a knockout of Mbnl1 and Mbnl2 during developmental expression of Ckm.

## CRISPR-mediated *mLimch1* deletion

Limch1 6exKO mice were developed in an FVB background in collaboration with the BCM mES Core. To delete the alternatively spliced exons 9–14 of Limch1, 4 single-guide RNAs (sgRNAs) were selected by the mES Core using the Wellcome Trust Sanger Institute Genome Editing website (http://www.sanger.ac.uk/htgt/wge/) positioned to flank the genomic region containing alternative exons 9–14 of Limch1 (5′ sgRNA: http://www.sanger.ac.uk/htgt/wge/crispr/485343919, http://www.sanger.ac.uk/htgt/wge/crispr/485343920 and 3′ sgRNA: http://www.sanger.ac.uk/htgt/wge/crispr/485345259, http://www.sanger.ac.uk/htgt/wge/crispr/485345265).

Briefly, fertilized oocytes were collected and microinjected with the sgRNA/CRISPR/Cas9 mixture in ~100 pronuclear stage zygotes. The injected zygotes were transferred into pseudopregnant females, at least 25 zygotes per recipient female. To confirm the null allele, F0 mice were genotyped by standard PCR targeting the flanking region outside the six exons. F1 animals from two F0 founders were used to start two independent lines. The junctions of the genomic deletions in both lines were determined by sequencing PCR products generated across the junctions. Both lines were backcrossed to WT FVB at least seven generations to out-cross off-target loci before performing experiments. Both lines showed comparable grip strength deficits, and one line was selected for experimentation.

## RT–PCR and RNA splicing

Quadriceps skeletal muscle tissues were dissected, flash-frozen in liquid nitrogen, and maintained at −80°C before RNA isolation. Total RNA was isolated from skeletal muscle tissues using TRIzol reagent (catalog number 15596018; Invitrogen), and cDNA was prepared from 1 μg of DNase-treated RNA (catalog number 4368813; Thermo Fisher Scientific) followed by PCR amplification. Human fetal skeletal muscle RNA was obtained from a commercial source (catalog number 1F60; Cell Applications), and human adult RNA was from autopsy samples from the National Disease Research Interchange (NDRI). cDNA was prepared from 1 μg of DNase-treated RNA followed by PCR amplification.

To analyze alternative splicing events, primers annealing to flanking constitutive exons were designed as follows: to amplify mouse uLimch1 (expected band size: 136 bp) and mLimch1 (expected band size: 103 bp) — F1: 5′-AGCGGGAGGAATTCAGGAAG-3′, R1: 5′-CGTCAATTCCCTCCTCCTCT-3′, R2: 5′-TCCTCACACCGCATGTCAAA-3′. To amplify total mouse Limch1 (expected band size: 74 bp) — F: 5′-

GAAGACGGTGAAGAAAGGCC-3′, R: 5′-GTTGTCGTAAAGTGCTGGGG-3′. To amplify human uLIMCH1 (expected band size: 344 bp) and mLIMCH1 (expected band size: 376 bp) — F1: 5′-ATGGTGAGCCGAAATCAGCA-3′, R1: 5′-CGCTTTTGGATCTCAGCAGC-3′, R2: 5′-CCAACGAGCCAGGT-CATCTT-3′. To amplify total human LIMCH1 (expected band size: 524 bp) — F: 5′-AAGATGACCTGGCTCGTTGG-3′, R: 5′-CTTGGGTGTCAG-CATAGGCA-3′. PCR products were resolved on a 5% polyacrylamide gel to determine both the total Limch1 RNA and the percentage of uLimch1 and mLimch1 RNA. Ethidium bromide stained RT–PCR bands were analyzed using Kodak Gel Logic 2000 and Carestream software. Percent spliced in (PSI) was calculated to determine the percent of mRNA from the gene that contains the alternative exons, quantified using densitometry according to the equation: $PSI = 100 \times$ (Inclusion band/[Inclusion band + Skipping band]). We obtained adult human DM1-affected and non-affected skeletal muscle autopsy samples from colleagues and the NDRI and performed RNA sequencing at Baylor College of Medicine.

## Protein isolation and immunoblotting

Tissue protein extracts were prepared from isolated skeletal muscle tissue by mechanical homogenization with a bullet blender (Next Advance) in 1X RIPA lysis buffer (catalog number 9806; Cell Signaling) supplemented with 1X Xpert Protease Inhibitor Cocktail Solution (catalog number P3100; GenDEPOT) with subsequent centrifugation to remove cell debris ($15,000g$ for 15 min at 4°C). BCA protein assay kit (catalog number PI23225; Thermo Fisher Scientific) was used to quantify protein concentration, and protein samples were diluted to 3 μg/μl in Laemmli SDS sample buffer (catalog number J61337.AC; Thermo Fisher Scientific) and boiled at 100°C for 5 min. 15–30 μg of protein lysate was loaded and separated on 12% Tris-glycine SDS-PAGE gels and transferred to nitrocellulose membranes (catalog number 1620167; Bio-Rad) using the Trans-Blot Turbo Transfer System (Bio-Rad) for Western blot analysis. Membranes were blocked with 5% milk in PSB-T (0.1% Tween-20; Sigma-Aldrich) for 1 h and incubated overnight at 4°C in 5% milk/PBS-T with the following antibodies: anti-LIMCH1 (1:400, catalog number HPA063840; Sigma-Aldrich), anti-MBNL1 (1:1,000, catalog number 3A4: sc-47740; Santa Cruz Biotechnology), anti-MBNL2 (1:1,000, catalog number 3B4: sc-136167; Santa Cruz Biotechnology), anti-GAPDH (1:10,000, catalog number 14C10: 2118; Cell Signaling), anti-histone H3 (1: 2,000, catalog number D1H2: 4499; Cell Signaling), and anti-mLIMCH1 (1:400; Biomatik). Membranes were washed three times with PBS-T and incubated for 1 h at room temperature in HRP-conjugated goat anti-rabbit or goat anti-mouse secondary antibody (1:10,000; Jackson Immunoresearch) in 5% milk/PBS-T after which membranes were washed three times with PBS-T. Immunoreactivity was detected using West-Q Pico Dura ECL Solution (Catalog number W3653; GenDEPOT), and membranes were imaged on a ChemiDoc XRS+ Imaging system (Bio-Rad).

## Grip strength and body weight

A grip strength meter (Columbus Instruments) was used to conduct an all-limb grip strength assessment on WT and Limch1 6exKO FVB mice at 10–12 wk of age. Mice were acclimated in the testing room for at least 10 min before testing to minimize the effects of stress on

behavior during testing. Mice genotypes were blinded to the experimenter, and the same experimenter conducted all grip strength assessments to reduce variability. As the mouse grasped the grid, a digital force transducer recorded the peak pull force in grams. The procedure was repeated for a total of three trials, separated by 15-min inter-trial intervals. At the end of trial three, animals were weighed, and grip force was standardized to the respective body weight of the individual mouse.

## Skeletal muscle weights

Muscle weights of the gastrocnemius, quadriceps, and soleus were measured on an analytical balance and normalized to tibia length at the conclusion of the grip strength assessment. Direct comparisons of normalized muscle weights were made between age-matched treatment groups.

## Histology

Quadriceps, soleus, and EDL skeletal muscle tissue were isolated, connective tissue removed, fixed in 10% formalin for 24 h, paraffin-embedded, and cut into 10-$\mu$m cross sections. Hematoxylin and eosin staining (H&E) and picrosirius staining of skeletal muscle were performed using standard procedures. Images were acquired using an Olympus BX41 microscope with an Olympus DP70 camera.

## Isometric ex vivo force analysis

Soleus and EDL muscles were dissected from male mice and using silk suture (4-0), one tendon of the muscle was tied to a fixed hook, and the other tendon of the muscle was tied to a force transducer (F30; Harvard Apparatus). The muscle was maintained in a physiological saline solution and supplemented with 95% $O_2$–5% $CO_2$ at 30°C. The optimal length of the muscle (Lo) was determined to produce maximum twitch force, whereas optimal voltage was determined after Lo. Muscles were allowed to equilibrate for 10 min after Lo and optimal voltage assessment. Pulse and train durations of 0.25 and 400 ms, respectively, were used to assess muscle contractile properties with current passing through two platinum electrodes (PanLab LE 12406; Harvard Apparatus). Kinetics, force–frequency, fatigue, and recovery from fatigue protocols were collected and analyzed in LabChart 8 (ADInstruments) with the Peak Analysis module. Force–frequency was tested by stimulating the muscle at 1, 5, 10, 20, 40, 60, 80, 120, 150, and 200 Hz with 1-min intervals between each frequency tested. After a 2-min rest period, the fatigue protocol was conducted by stimulating the soleus with a 40 Hz stimulus and the EDL with a 70 Hz stimulus every 2 s for 5 min. After a 5-min rest period, the recovery from fatigue protocol was assessed by stimulating the muscle with alternating frequencies of 40 and 150 Hz every minute for 10 min. At the end of the recovery protocol, muscle length was measured using a hand-held electronic caliper, muscles were removed from the connected apparatus, excess connective tissue was removed, and muscles were blotted dry and weighed. For force–frequency, the muscle weight and Lo recorded were used to calculate absolute force expressed as N/cm² (Close, 1972). The rate of contraction and rate of relaxation were collected during the force–frequency protocol during a single

twitch (1 Hz) and was based on the slope of the force curve. The muscle weight and Lo recorded were used to determine the forces expressed as a function of time (N/cm²/s) for the rate of recovery (−dp/dt) and rate of relaxation (dp/dt). For the fatigue protocol, the force was normalized to the first stimulus before fatigue. For the recovery protocol, the force was normalized to a 40 or 150 Hz stimulus before fatigue.

## FDB myofiber isolation

Mice were anesthetized with 2% isoflurane and euthanized with subsequent cervical dislocation. FDB tissue was surgically dissected, and connective tissue was removed. The FDB muscle was placed into a 35-mm dish containing minimum essential medium (catalog number 51411C; Sigma-Aldrich), 2 mg/ml collagenase type 1 (catalog number C0130; Sigma-Aldrich), 0.1% gentamycin, and incubated in 5% $CO_2$ at 37°C for 2 h. After incubation, the FDB muscle was washed with PBS, transferred to a new, non-coated 35-mm dish, and single fibers were gently released with serial trituration into 2 ml of DMEM (catalog number 11-965-092; Thermo Fisher Scientific) supplemented with 10% FBS (catalog number F0900-050; GenDEPOT), 1% penicillin/streptomycin, and incubated in 5% $CO_2$ at 37°C overnight. The isolated myofibers were then used for subsequent assays.

## Cell culture and in vitro transfections

C2C12 cells were cultured in 96-well plates (catalog number 89626; Ibidi) pre-coated with ECM gel (catalog number E1270; Sigma-Aldrich) and maintained in DMEM supplemented with 10% FBS, 1% penicillin/streptomycin, and incubated in 5% $CO_2$ at 37°C. Once C2C12 cells reached 70–80% confluency, cells were transfected with *uLimch1* and *mLimch1* epitope-tagged expression plasmids using Lipofectamine 3000 following the manufacturer's protocol (catalog number L3000015; Invitrogen). *mLimch1* and *uLimch1* epitope-tagged expression plasmids were commercially synthesized by GenScript. A Myc tag was included in the C-terminus region of the *uLimch1* sequence, whereas a Myc tag was included in the C-terminus region of the *mLimch1* sequence in addition to a FLAG tag which was inserted within the muscle-specific sequence of *mLimch1* in exon 10. The *uLimch1* and *mLimch1* sequences were cloned into the backbone of a pcDNA3.1 plasmid by GenScript. Transfected cells were incubated in antibiotic-free DMEM supplemented with 10% FBS for 48 h. Cells were then fixed in 10% formalin for 10 min and prepared for immunofluorescence and imaging.

## mLIMCH1 antibody synthesis

The mLIMCH1-specific primary antibody was commercially generated by Biomatik. The following protein antigen sequence was submitted to Biomatik for antibody synthesis-LEQAGIKVMPAAQRF ASQKQLSEEKEAIRDIVLRKENSFLTHQHGNDSEAEGEVVCRLPDLEKDDFAA RRARMNQTKPMVPLNQLLYGPY. Rabbits underwent multiple immunizations with the antigen, which was expressed as a recombinant protein, to yield an immunogenic response. After multiple immunizations, the serum titer reached at least 1:8,000, and 3–5 mg of

antibody was purified with antigen affinity purification. The resulting antibody was validated by Biomatik with ELISA (>1:32,000) and validated for Western blot and immunofluorescence by our laboratory.

## Immunofluorescence

96-well plates (Ibidi) were coated with ECM gel for 30 min at room temperature. 100 $\mu$l of isolated myofibers were transferred into the coated wells and incubated in 5% $CO_2$ at 37°C for 1 h to allow for myofiber adhesion. Myofibers were washed with PBS and fixed with 4% paraformaldehyde in PBS for 10 min at room temperature. Fixed myofibers were washed with PBS three times and permeabilized using 0.1% Triton X-100 (Sigma-Aldrich) in PBS for 10 min. Myofibers were then blocked in 5% FBS, 0.1% Triton X-100 in PBS at room temperature for 1 h and incubated overnight in 5% FBS and 0.1% Triton X-100 in PBS at 4°C with the following antibodies: anti-LIMCH1 (1:200, catalog number HPA063840; Sigma-Aldrich), anti-mLIMCH1 (1:200; Biomatik), anti-RYR1 (1:200, catalog number 34C; Developmental Studies Hybridoma Bank), and anti-BIN1 (1:200, catalog number 14647-1-AP; Proteintech). Transfected C2C12 cells were fixed and stained following the same protocol and incubated overnight in 5% FBS and 0.1% Triton X-100 in PBS at 4°C with the following antibodies: anti-mLIMCH1 (1:200; Biomatik), anti-Myc (1:200, 71D10: 2278; Cell Signaling), and anti-DDDDK (1:200, catalog number ab1257; Abcam). The following day, the myofibers or cells were washed three times with PBS and incubated in goat anti-rabbit 555 Alexa Fluor–conjugated secondary antibody (1:1,000, catalog number A27039; Thermo Fisher Scientific) or goat anti-mouse 488 Alexa Fluor–conjugated secondary antibody (1:1,000, catalog number A11029; Thermo Fisher Scientific) for 2 h in 5% FBS and 0.1% Triton X-100, and washed three times with PBS. Transfected C2C12 cells were stained with DAPI (0.5 $\mu$g/ml, catalog number D9542; Sigma-Aldrich) for 5 min, followed by one wash with PBS before imaging. For T-tubule staining with FM 4–64 (Catalog number T13320; Invitrogen), unfixed myofibers were loaded with 5 $\mu$g/ml of FM 4–64 dye and imaged immediately.

## Microscopy

Confocal microscopy on myofibers was performed using a Nikon A1-Rs inverted laser scanning microscope with a 60× Plan Apo VC/1.4 NA oil-immersion objective. Transfected C2C12 images were acquired using DeltaVision Elite (GE Healthcare) and processed using SoftWoRx software (GE Healthcare).

## Image processing and quantitative analysis

Plot profiles and image analytics were performed using Fiji (ImageJ). A mask was generated, delimiting each myofiber by applying a threshold manually. The identified signal was then projected onto the original image file outlining the area for analysis. In the case of LIMCH1 immunofluorescence, the SD of the immunofluorescence signal was determined to measure the signal distribution of LIMCH1 throughout the entire myofiber of a single image plane. Two horizontal slices were chosen near the center of the myofiber, and the SD of immunofluorescence from each slice was averaged for each sample analyzed. Multiple myofibers from one mouse were analyzed and averaged to serve as a single sample, with six mice (n = 6) used for both the WT and *Limch1* 6exKO groups. Representative plot profiles were recorded from HOM *Limch1* 6exKO and WT myofibers through the myofiber (sarcolemma to sarcolemma) and along the sarcolemma in the center plane of the myofiber.

## Cellular Ca²⁺imaging

FDB muscle was surgically dissected, and individual myofibers were isolated as described above. 22 × 22-mm micro cover glass (catalog number 48366227; VWR) was coated with ECM gel for 30 min at room temperature. 100 $\mu$l of isolated myofibers were transferred onto the coated cover glass slides and incubated in 5% $CO_2$ at 37°C for 30 min to allow for myofiber adhesion. Once myofibers adhered, physiological saline solution (1.8 mM $CaCl_2$, 120 mM NaCl, 4.7 mM KCl, 0.6 mM $MgSO_4$, 1.6 mM $NaHCO_3$, 0.13 mM $KH2PO_4$, 7.8 mM glucose, 20 mM HEPES, pH 7.3) was added in a buffer exchange, and 5 $\mu$M Fura-4F AM (catalog number F14175; Invitrogen) was added to the myofibers for 30 min at room temperature. Myofibers were washed with physiological saline solution at least three times to remove Fura-4F AM and allowed to de-esterify for 30 min. To determine optimal calcium response, the focal plane was adjusted, and calcium transients were measured following twitch (1 Hz), 20 Hz tetanus, and 100 Hz tetanus with pulse and train durations of 0.5 and 250 ms, respectively. IonOptix Myocyte Calcium and Contractility Recording System (IonOptix) was used to monitor Fura-4F excitation (360/380 nm) and emission (510 nm). Intracellular calcium changes during myofiber stimulation were inferred from the recorded 360 nm/380 nm ratio signals. Multiple myofibers from one mouse were analyzed and averaged to serve as a single sample, with each sample (n) derived from different mice.

## Statistics

All quantitative experiments have at least three independent biological replicates with the exception of the RT–PCR of Ckm Cre; *Mbnl1*ᶠˡ/ᶠˡ, *Mbnl2*ᶠˡ/ᶠˡ, and WT skeletal muscle. Results are presented as mean ± SEM. All statistical analyses were calculated using Prism software (version 8.0; GraphPad), and the statistical method and sample sizes used are shown in respective figure legends. *P*-values were calculated by a two-tailed *t* test for all the experiments, and a *P*-value of less than 0.05 was considered statistically significant.

## Study approval

All mouse experiments were carried out in accordance with the *Guide for the Care and Use of Laboratory Animals* (National Academies Press, 2011) and approved by the Baylor College of Medicine Institutional Animal Care and Use Committee.

# Supplementary Information

# Acknowledgements

We thank Ashish Rao for his valuable input on experimental design and manuscript review. We also thank Riya Thomas and Jennifer Graham for their help with the grip strength assessment. We also thank members of the Cooper laboratory for their valuable discussions and help throughout the project. Imaging for this project was supported by the Integrated Microscopy Core at Baylor College of Medicine and the Center for Advanced Microscopy and Image Informatics (CAMII) with funding from the NIH (DK56338, CA125123, ES030285), and CPRIT (RP150578, RP170719) with a special thanks to Hannah Johnson and Fabio Stossi for their assistance and guidance. This project was supported by the Mouse Metabolism and Phenotyping Core at Baylor College of Medicine, which is subsidized as a member of the institution's Advanced Technology Core and through funding from the NIH (UM1HG006348, R01DK114356, R01HL130249). This project was supported by the Genetically Engineered Rodent Model (GERM) Core at BCM. The GERM Core is funded in part by the National Institutes of Health Cancer Center Grant (P30 CA125123). Images were created with BioRender. This project was supported by NIH grants R01 AR060733 (TA Cooper), R01 HL147020 (TA Cooper), F31AR078646 (MS Penna), R01 AR061370 (GG Rodney), the Myotonic Dystrophy Foundation fellowship (R-C Hu), and by generous support from the Baylor Research Advocates for Student Scientists (BRASS) who provided financial support (MS Penna) for many of the experiments conducted in this study.

## Author Contributions

MS Penna: conceptualization, data curation, formal analysis, funding acquisition, validation, investigation, methodology, and writing—original draft, review, and editing.
R-C Hu: data curation, formal analysis, validation, investigation, methodology, and writing—original draft, review, and editing.
GG Rodney: conceptualization, resources, supervision, investigation, methodology, and writing—original draft, review, and editing.
TA Cooper: conceptualization, resources, software, formal analysis, supervision, funding acquisition, methodology, and writing—original draft, review, and editing.

## Conflict of Interest Statement

The authors declare that they have no conflict of interest.

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
