## [Reviewer comments · Life Science Alliance]

Life Science Alliance

The role of Limch1 alternative splicing in skeletal muscle function

Matthew Penna, Rong-Chi Hu, George Rodney, and Thomas Cooper

DOI: <https://doi.org/10.26508/lsa.202201868>

Corresponding author(s): Thomas Cooper, Baylor College of Medicine and George Rodney, Baylor College of Medicine

Review Timeline:

Submission Date:	2022-12-07
Editorial Decision:	2022-12-08
Revision Received:	2023-01-28
Editorial Decision:	2023-03-06
Revision Received:	2023-03-13
Accepted:	2023-03-15

Transaction Report:

Please note that the manuscript was reviewed at Review Commons and these reports were taken into account in the decision-making process at Life Science Alliance.

Manuscript number: RC-2022- 01699R

Corresponding author(s): Thomas A. Cooper

[The “revision plan” should delineate the revisions that authors intend to carry out in response to the points raised by the referees. It also provides the authors with the opportunity to explain their view of the paper and of the referee reports.]

The document is important for the editors of affiliate journals when they make a first decision on the transferred manuscript. It will also be useful to readers of the reprint and help them to obtain a balanced view of the paper.

*If you wish to submit a full revision, please use our "Full Revision" template. **It is important to use the appropriate template to clearly inform the editors of your intentions.**]*

1. General Statements

We thank the reviewers from Review Commons for their thorough reviews of our manuscript entitled, “The role of *Limch1* alternative splicing in skeletal muscle function.” We were delighted by the many supportive comments of all three reviewers calling our study a “definite advance in our understanding of developmentally-regulated splice isoform transitions that are disease relevant”, “good comprehensive study with convincing results, the design of the experiments is good, and the conclusions are solid”, and “The article is well written, and I favor the publication of this [article] with minor revisions.”

The reviews include comments on the interest in identifying the mechanism of action of mLIMCH1 in skeletal muscle function such as “ [The study] presents multiple new tools to study mLimch1 and identifies a possible role for mLIMCH1 in calcium regulation, but stops short of identifying the mechanism by which this regulation occurs.” While we agree that how the skeletal muscle-specific isoform of LIMCH1 affects calcium handling is of interest, we respectfully suggest that this manuscript describe previously unknown biology that will be of interest to investigators in different fields including muscle physiology, alternative splicing regulation, and skeletal muscle pathology in myotonic dystrophy. All experiments in this manuscript are performed in vivo using skeletal muscle tissues from animals lacking the isoform of *Limch1* that is expressed only in skeletal muscle and is normally induced after birth. Comparisons were made to age-matched wild-type control animals, often litter mates. The results establish the functional significance of the LIMCH1 protein and particularly the muscle-specific isoform in skeletal muscle through extensive analysis of LIMCH1 localization and the impact of mLIMCH1 knockout on muscle strength, force generation, calcium handling and the disease relevance of this splicing transition in myotonic dystrophy type 1. Please review the comments of all three reviewers who were quite favorable to the significance of the work and overall favorable to its publication. Below, we clarify and describe additional data that has, and will be

added to the manuscript to address all comments of the reviewers. Our responses are in blue font.

2. Description of the planned revisions

Reviewer 1

- “Page 6 - data not shown. The point of conservation is not essential to this story, but the authors should either include a table or panel with that data, or remove the data not shown statement. Given the putative relevance to DM1, it might be preferable to include data to support the developmental transition in human data.” We have removed the “data not shown” statement as suggested and we highlighted the importance of conservation of the induction of a skeletal muscle isoform of LIMCH1 after birth as a strong indication of functional importance for the isoform. We agree that data showing the conserved LIMCH1 splicing transition in human skeletal muscle development will support this point. We will include RT-PCR analysis of LIMCH1 splicing in fetal and adult human skeletal muscle RNA in Figure 6 to support the reversion of splicing to the fetal pattern observed in DM1. The results will complement the normal *Limch1* splicing transition in mice (Figure 1) and the normal and aberrant fetal splicing patterns shown for unaffected and DM1 adult skeletal muscle, respectively (Figure 6).

3. Description of the revisions that have already been incorporated in the transferred manuscript

Please insert a point-by-point reply describing the revisions that were already carried out and included in the transferred manuscript. If no revisions have been carried out yet, please leave this section empty.

Reviewer 1

- “Figure 4 - The authors do a nice experiment to show the localization of Limch1 and raise an antibody to detect the muscle specific isoform. The data seem to show that the muscle-specific isoform localizes to the sarcolemma, and this staining is largely lost in the mutant mice. **By contrast, one could infer that the cytoplasmic signal in the WT comes from the ubiquitous isoform (which accounts for 30-40% of the Limch1 expression).** This is consistent with the validation in Fig. 2. However, the authors in the text claim this experiment reveals an increased distribution throughout the myofiber, or a more even signal distribution in the cytoplasm, and that the uLimch1 cannot recapitulate mLimch1 localization. Fig. 2 suggests that total levels of Limch1 are increased (as noted by the authors in the discussion). **Given that the muscle-specific isoform localizes to the sarcolemma, and the ubiquitous isoform is presumably sarcoplasmic, it isn't clear to me that there is any change in localization per se.** What the authors show is just that the signal at the sarcolemma is lost, and if one compares the intensity in the right-hand plots in Fig. 4B, they are comparable in the sarcoplasmic region. It seems likely there is more of the ubiquitous isoform, and what is seen here is just how that isoform localizes. The quantification the authors

perform in D would likely show this strong difference in the localization of the muscle isoform. If the authors redo this quantification, exclude the signal at the sarcolemma and normalize to the average pixel intensity in the fiber, do they still see a difference? I am not convinced that the "clustering" of the signal of the ubiquitous, cytoplasmic isoform is in any way changed. Given the difference in the two proteins, I also would not expect that the ubiquitous isoform could compensate for loss of the muscle isoform, and would not expect it to "recapitulate" the muscle-isoform localization." We agree that Figure 4 and the explanation in the text was not clear and we thank the reviewer for pointing this out. We have addressed this concern by modifying the figure as suggested by the reviewer and clarifying the description in the results section. The main point, that is recognized by the reviewer but needed clarification, is that the mLIMCH1 isoform preferentially localizes to the sarcolemma and the uLIMCH1 isoform is preferentially cytoplasmic. In the HOM *Limch1* 6exKO myofibers, the increased cytoplasmic signal is due to the increased level of uLIMCH1 as shown by the western blot in Figure 2. The reviewer is correct that there is not a "change in localization of isoforms per se". We clarified this point to highlight the differential localization of the uLIMCH1 and mLIMCH1 isoforms within the sarcolemma vs. the sarcoplasm. The revision of the plot profile in Figure 4B and the analysis of the standard deviation of signal in Figure 4D demonstrates the stark difference in staining observed between the HOM *Limch1* 6exKO and WT myofibers when stained with a pan-LIMCH1 antibody. The signal intensity plot profile from sarcolemma to sarcolemma (Figure 4B) indicates that the uLIMCH1 isoform is not "mis-localized" upon mLIMCH1 knockout as we originally (mis)-stated. Upon mLIMCH1 knockout, there is increased uLIMCH1 expression compared to WT myofibers. Considering this in combination with the sarcolemma preference of mLIMCH1 (Figure 4E) and the significant loss of signal in the sarcolemma region in *Limch1* 6exKO myofibers, we conclude that in HOM *Limch1* 6exKO myofibers, uLIMCH1 is primarily localized throughout the sarcoplasm.

Reviewer 1 (optional)

- "Experiments looking more closely at LIMCH1 co-localization with other proteins at the sarcolemma or the sufficiency of the muscle-specific region to localize would also be useful (for example, can the muscle-specific region localize GFP to the membrane in cells?)." We performed immunofluorescence microscopy of LIMCH1 with several skeletal muscle-relevant proteins but did not observe: (1) disruptions of normal structures in HOM *Limch1* 6exKO compared to WT myofibers or (2) colocalization that helped clarify any mechanistic role of mLIMCH1 or uLIMCH1. Therefore this data was not included in the original manuscript. In regard to the suggestion on the sufficiency of the muscle-specific region to localize to the sarcolemma region, we had previously generated a plasmid to express a fluorescent protein fused to the protein encoded by the six skeletal muscle-specific exons of LIMCH1 but it failed to localize to the sarcolemma. In collaboration with protein structural experts at Baylor College of Medicine, we analyzed the skeletal muscle-specific region of LIMCH1 and found it to be entirely disordered without known homologs. It appears that this region has no secondary structure but when expressed within the entire LIMCH1 protein

which has conserved domains (calponin homology, LIM, coiled-coil regions) and upon protein binding, it is possible for the region to adopt a structure facilitating its binding in the sarcolemma region. Therefore we believe that regions common to both isoforms are required in combination with the muscle-specific region for preferential localization to the sarcolemma.

Reviewer 1 (minor comments)

- “In the Figure 3 legend, the order of the descriptions for B-C and D-E is switched. The order of the panels matches the text, but the legend switches the description of the force-frequency curves (shown in B & C but labeled as D & E), with the description of the rate of relaxation and contraction plots (shown in D and E but labeled as B and C in the legend).” We fixed this error and thank the reviewer for pointing it out.
- “The scale in Figure 4, panel B between the top and bottom plots is not the same, so it is difficult to compare, particularly for the panels on the right. See comment above.” In addition to clarifying uLIMCH1’s localization upon mLIMCH1 knockout within the text, we added figure titles above the plot profile which will clarify the different plot profiles for the reader. In regard to the comment about the scale of the plot profile, we have addressed this by re-scaling the two plot profiles on the right in Figure 4B. These plot profiles now share the same scale, which is advantageous because this plot profile better emphasizes the stark difference in signal observed between the sarcolemma and sarcoplasm in WT myofibers that is lacking in HOM *Limch1* 6exKO myofibers.

Reviewer 2

- “Figure 6A: There is a discrepancy between gene structures and splicing isoforms shown in Fig. 1 vs Fig. 6. There are differences in spacing between exons, and there appear to be six exons in the differentially regulated region in Fig 1, but seven exons in Fig 6. Perhaps this is a difference between human and mouse genes? Does the human gene actually regulate seven exons in this region, rather than six exons in the mouse? In both figures the gene is labeled as *Limchi1*, and both figures indicate that the ubiquitous isoform lacks exons 9-14. Please clarify.” The reviewer is correct that the human mLIMCH1 isoform contains seven exons that are skeletal muscle-specific compared with the six exons that are skeletal muscle-specific in the mouse. The seven human exons encode 544 amino acids with 65% homology with the mouse segment. We have clarified this in the figure legend and text. Exons 9-14 are shown in Figure 6B since this diagrams the mouse gene.
- “The methods section on RT-qPCR and RNA splicing presumably refers to analysis of mouse tissues. What is the origin of the human DM1 RNA-seq data?” We obtained adult human DM1-affected and non-affected skeletal muscle autopsy samples from colleagues and the NDRI and performed RNA-sequencing at Baylor College of Medicine. The RNA-seq has not yet been published, but we include the data for *LIMCH1* to demonstrate the dramatic change in the

alternative splicing pattern in DM1 skeletal muscle tissue. This has been clarified in the methods section.

- “Perhaps the word "activity" should be deleted in the following sentence: "The sole study investigating the function of LIMCH1 characterized it as an actin stress fiber associated protein that binds non-muscle myosin 2A (NM2A) activity to regulate focal adhesion formation." We thank the reviewer for pointing this out and we have removed this word.

Reviewer 3

- “The diminution of the muscle force production in *Limch16exKO* is not correlated with a change in morphology of the myofibers in H&E and picosirius stainings (Fig S2). Did the authors look at other skeletal muscles, fiber type, size, or different time points? (The age of the mouse and the name of the skeletal muscle used for the histology could be included in the results sections or figure legend).” As suggested by Reviewer 3, we have included additional histological data in Supplementary Figure 2. In addition to the histology at 10-12 weeks of age, the new data includes histology of multiple skeletal muscle tissues (quadriceps, EDL, soleus) at one year of age. The histology of *Limch1 6exKO* tissue at different time points showed no morphological differences (centralized nuclei or fibrosis) consistent with no change in muscle weight which led us to emphasize the significant effect of mLIMCH1 knockout on skeletal muscle function in the absence of muscle loss or overt structural changes. In regard to fiber-type, we have included histology of both the EDL (fast-twitch) and soleus (slow-twitch) and even after one year, we observe no gross morphological differences. Additionally, we analyzed the force production of both the EDL and soleus (Figure 3) with the fiber-type predominance of these tissues in mind and found decreased force generation in both tissues. We included the types of skeletal muscle tissue analyzed and the age of the mice in Supplementary Figure 2 as per the reviewer’s suggestion.
- “The authors performed RNAseq analysis in the skeletal muscle of the KO mouse (Fig 2B). What is the result of this experiment? Is the KO muscle transcriptome different or similar to control muscles?” We conducted RNA-sequencing on tissue from HOM *Limch1 6exKO* and WT controls and the results were disappointing showing minor differences that did not contribute to understanding the phenotype. We used this data only to show the loss of the six exons in Fig. 2B, however, we decided that RT-PCR analysis was the better assay since it shows not only that the exons are not included but also that exons 8 and 15 are spliced correctly, which is not apparent using the RNA-seq displayed on the genome browser.

4. Description of analyses that authors prefer not to carry out

Please include a point-by-point response explaining why some of the requested data or additional analyses might not be necessary or cannot be provided within the scope of a

revision. This can be due to time or resource limitations or in case of disagreement about the necessity of such additional data given the scope of the study. Please leave empty if not applicable.

Reviewer 1 (Both points listed as optional)

- “If the authors perform TEM, can they see defects in t-tubules or organization of the sarcoplasmic reticulum, that are not visible by light microscopy?” We considered conducting TEM to investigate sarcomeric, T-tubule, or sarcolemma changes in myofibers derived from HOM *Limch1* 6exKO mice, but we concluded that it would most likely be of limited use. We do not think that T-tubule structural changes will be observed via TEM primarily due to the challenges of finding significant changes compared to WT controls in which one can always find abnormal structures. In our experience and the experience of our collaborator (Dr. Rodney) the disruptions must be dramatic to distinguish from the noncanonical structures often observed. Thus, we do not plan on conducting TEM to identify defects in the T-tubules.
- “If the muscle-specific isoform is transfected or transduced into differentiated myotubes, how does this affect calcium dynamics in the culture system?” While an interesting idea, we do not plan on conducting this experiment for multiple reasons. One issue is that all of our data is derived from in vivo analysis or from isolated myofibers and our concern is that the relatively immature state of myotubes in culture will provide a poor comparison to isolated myofibers. Therefore, we believe that it will be difficult to add meaningful data to the calcium data presented in Figure 5 through this experiment. Additionally, we have observed mis-localization of the overexpressed uLIMCH1 and mLIMCH1 in C2C12 cells that we believe would add too many caveats for meaningful interpretation of the results, regardless of the effects on calcium dynamics

December 8, 2022

Re: Life Science Alliance manuscript #LSA-2022-01868-T

Dr. Thomas Cooper
Baylor College of Medicine

Dear Dr. Cooper,

Thank you for submitting your manuscript entitled "The role of Limch1 alternative splicing in skeletal muscle function" to Life Science Alliance. We invite you to re-submit the manuscript, revised according to your Revision Plan.

Thank you for this interesting contribution to Life Science Alliance. We are looking forward to receiving your revised manuscript.

Sincerely,

B. MANUSCRIPT ORGANIZATION AND FORMATTING:

[The “revision plan” should delineate the revisions that authors intend to carry out in response to the points raised by the referees. It also provides the authors with the opportunity to explain their view of the paper and of the referee reports.

The document is important for the editors of affiliate journals when they make a first decision on the transferred manuscript. It will also be useful to readers of the reprint and help them to obtain a balanced view of the paper.

*If you wish to submit a full revision, please use our "Full Revision" template. **It is important to use the appropriate template to clearly inform the editors of your intentions.**]*

1. General Statements

We thank the reviewers from Review Commons for their thorough reviews of our manuscript entitled, “The role of *Limch1* alternative splicing in skeletal muscle function.” We were delighted by the many supportive comments of all three reviewers calling our study a “definite advance in our understanding of developmentally-regulated splice isoform transitions that are disease relevant”, “good comprehensive study with convincing results, the design of the experiments is good, and the conclusions are solid”, and “The article is well written, and I favor the publication of this [article] with minor revisions.”

The reviews include comments on the interest in identifying the mechanism of action of mLIMCH1 in skeletal muscle function such as “ [The study] presents multiple new tools to study mLimch1 and identifies a possible role for mLIMCH1 in calcium regulation, but stops short of identifying the mechanism by which this regulation occurs.” While we agree that how the skeletal muscle-specific isoform of LIMCH1 affects calcium handling is of interest, we respectfully suggest that this manuscript describe previously unknown biology that will be of interest to investigators in different fields including muscle physiology, alternative splicing regulation, and skeletal muscle pathology in myotonic dystrophy. All experiments in this manuscript are performed in vivo using skeletal muscle tissues from animals lacking the isoform of *Limch1* that is expressed only in skeletal muscle and is normally induced after birth. Comparisons were made to age-matched wild-type control animals, often litter mates. The results establish the functional significance of the LIMCH1 protein and particularly the muscle-specific isoform in skeletal muscle through extensive analysis of LIMCH1 localization and the impact of mLIMCH1 knockout on muscle strength, force generation, calcium handling and the disease relevance of this splicing transition in myotonic dystrophy type 1. Please review the comments of all three reviewers who were quite favorable to the significance of the work and overall favorable to its publication. Below, we clarify and describe additional

Revision Plan

data that has, and will be added to the manuscript to address all comments of the reviewers. Our responses are in blue font.

2. Description of the planned revisions

- All planned revisions have been included in the updated manuscript.

3. Description of the revisions that have already been incorporated in the transferred manuscript

Please insert a point-by-point reply describing the revisions that were already carried out and included in the transferred manuscript. If no revisions have been carried out yet, please leave this section empty.

Reviewer 1

- “Page 6 - data not shown. The point of conservation is not essential to this story, but the authors should either include a table or panel with that data, or remove the data not shown statement. Given the putative relevance to DM1, it might be preferable to include data to support the developmental transition in human data.” We agree that data showing the conserved LIMCH1 splicing transition in human skeletal muscle development will support this point. We included RT-PCR analysis of LIMCH1 splicing in fetal and adult human skeletal muscle RNA in Figure 1 to support the reversion of splicing to the fetal pattern observed in DM1. The results will complement the normal *Limch1* splicing transition in mice (Figure 1) and the normal and aberrant fetal splicing patterns shown for unaffected and DM1 adult skeletal muscle, respectively (Figure 6).
- “Figure 4 - The authors do a nice experiment to show the localization of *Limch1* and raise an antibody to detect the muscle specific isoform. The data seem to show that the muscle-specific isoform localizes to the sarcolemma, and this staining is largely lost in the mutant mice. **By contrast, one could infer that the cytoplasmic signal in the WT comes from the ubiquitous isoform (which accounts for 30-40% of the *Limch1* expression).** This is consistent with the validation in Fig. 2. However, the authors in the text claim this experiment reveals an increased distribution throughout the myofiber, or a more even signal distribution in the cytoplasm, and that the u*Limch1* cannot recapitulate m*Limch1* localization. Fig. 2 suggests that total levels of *Limch1* are increased (as noted by the authors in the discussion). **Given that the muscle-specific isoform localizes to the sarcolemma, and the ubiquitous isoform is presumably sarcoplasmic, it isn't clear to me that there is any change in localization per se.** What the authors show is just that the signal at the sarcolemma is lost, and if one compares the intensity in the right-hand plots in Fig. 4B, they are comparable in the sarcoplasmic region. It seems likely there is more of the ubiquitous isoform, and what is seen here is just how that isoform localizes. The quantification the authors perform in D would likely show this strong difference in the localization of the muscle isoform. If the authors redo this quantification,

Revision Plan

exclude the signal at the sarcolemma and normalize to the average pixel intensity in the fiber, do they still see a difference? I am not convinced that the "clustering" of the signal of the ubiquitous, cytoplasmic isoform is in any way changed. Given the difference in the two proteins, I also would not expect that the ubiquitous isoform could compensate for loss of the muscle isoform, and would not expect it to "recapitulate" the muscle-isoform localization." We agree that Figure 4 and the explanation in the text was not clear and we thank the reviewer for pointing this out. We have addressed this concern by modifying the figure as suggested by the reviewer and clarifying the description in the results section. The main point, that is recognized by the reviewer but needed clarification, is that the mLIMCH1 isoform preferentially localizes to the sarcolemma and the uLIMCH1 isoform is preferentially cytoplasmic. In the HOM *Limch1* 6exKO myofibers, the increased cytoplasmic signal is due to the increased level of uLIMCH1 as shown by the western blot in Figure 2. The reviewer is correct that there is not a "change in localization of isoforms per se". We clarified this point to highlight the differential localization of the uLIMCH1 and mLIMCH1 isoforms within the sarcolemma vs. the sarcoplasm. The revision of the plot profile in Figure 4B and the analysis of the standard deviation of signal in Figure 4D demonstrates the stark difference in staining observed between the HOM *Limch1* 6exKO and WT myofibers when stained with a pan-LIMCH1 antibody. The signal intensity plot profile from sarcolemma to sarcolemma (Figure 4B) indicates that the uLIMCH1 isoform is not "mis-localized" upon mLIMCH1 knockout as we originally (mis)-stated. Upon mLIMCH1 knockout, there is increased uLIMCH1 expression compared to WT myofibers. Considering this in combination with the sarcolemma preference of mLIMCH1 (Figure 4E) and the significant loss of signal in the sarcolemma region in *Limch1* 6exKO myofibers, we conclude that in HOM *Limch1* 6exKO myofibers, uLIMCH1 is primarily localized throughout the sarcoplasm.

Reviewer 1 (optional)

- "Experiments looking more closely at LIMCH1 co-localization with other proteins at the sarcolemma or the sufficiency of the muscle-specific region to localize would also be useful (for example, can the muscle-specific region localize GFP to the membrane in cells?)." We performed immunofluorescence microscopy of LIMCH1 with several skeletal muscle-relevant proteins but did not observe: (1) disruptions of normal structures in HOM *Limch1* 6exKO compared to WT myofibers or (2) colocalization that helped clarify any mechanistic role of mLIMCH1 or uLIMCH1. Therefore this data was not included in the original manuscript. In regard to the suggestion on the sufficiency of the muscle-specific region to localize to the sarcolemma region, we had previously generated a plasmid to express a fluorescent protein fused to the protein encoded by the six skeletal muscle-specific exons of LIMCH1 but it failed to localize to the sarcolemma. In collaboration with protein structural experts at Baylor College of Medicine, we analyzed the skeletal muscle-specific region of LIMCH1 and found it to be entirely disordered without known homologs. It appears that this region has no secondary structure but when expressed within the entire LIMCH1 protein

Revision Plan

which has conserved domains (calponin homology, LIM, coiled-coil regions) and upon protein binding, it is possible for the region to adopt a structure facilitating its binding in the sarcolemma region. Therefore we believe that regions common to both isoforms are required in combination with the muscle-specific region for preferential localization to the sarcolemma.

Reviewer 1 (minor comments)

- “In the Figure 3 legend, the order of the descriptions for B-C and D-E is switched. The order of the panels matches the text, but the legend switches the description of the force-frequency curves (shown in B & C but labeled as D & E), with the description of the rate of relaxation and contraction plots (shown in D and E but labeled as B and C in the legend).” We fixed this error and thank the reviewer for pointing it out.
- “The scale in Figure 4, panel B between the top and bottom plots is not the same, so it is difficult to compare, particularly for the panels on the right. See comment above.” In addition to clarifying uLIMCH1’s localization upon mLIMCH1 knockout within the text, we added figure titles above the plot profile which will clarify the different plot profiles for the reader. In regard to the comment about the scale of the plot profile, we have addressed this by re-scaling the two plot profiles on the right in Figure 4B. These plot profiles now share the same scale, which is advantageous because this plot profile better emphasizes the stark difference in signal observed between the sarcolemma and sarcoplasm in WT myofibers that is lacking in HOM *Limch1* 6exKO myofibers.

Reviewer 2

- “Figure 6A: There is a discrepancy between gene structures and splicing isoforms shown in Fig. 1 vs Fig. 6. There are differences in spacing between exons, and there appear to be six exons in the differentially regulated region in Fig 1, but seven exons in Fig 6. Perhaps this is a difference between human and mouse genes? Does the human gene actually regulate seven exons in this region, rather than six exons in the mouse? In both figures the gene is labeled as *Limchi1*, and both figures indicate that the ubiquitous isoform lacks exons 9-14. Please clarify.” The reviewer is correct that the human mLIMCH1 isoform contains seven exons that are skeletal muscle-specific compared with the six exons that are skeletal muscle-specific in the mouse. The seven human exons encode 544 amino acids with 65% homology with the mouse segment. We have clarified this in the figure legend and text. Exons 9-14 are shown in Figure 6B since this diagrams the mouse gene.
- “The methods section on RT-qPCR and RNA splicing presumably refers to analysis of mouse tissues. What is the origin of the human DM1 RNA-seq data?” We obtained adult human DM1-affected and non-affected skeletal muscle autopsy samples from colleagues and the NDRI and performed RNA-sequencing at Baylor College of Medicine. The RNA-seq has not yet been published, but we include the data for *LIMCH1* to demonstrate the dramatic change in the

Revision Plan

alternative splicing pattern in DM1 skeletal muscle tissue. This has been clarified in the methods section.

- “Perhaps the word "activity" should be deleted in the following sentence: "The sole study investigating the function of LIMCH1 characterized it as an actin stress fiber associated protein that binds non-muscle myosin 2A (NM2A) activity to regulate focal adhesion formation." We thank the reviewer for pointing this out and we have removed this word.

Reviewer 3

- “The diminution of the muscle force production in *Limch16exKO* is not correlated with a change in morphology of the myofibers in H&E and picosirius stainings (Fig S2). Did the authors look at other skeletal muscles, fiber type, size, or different time points? (The age of the mouse and the name of the skeletal muscle used for the histology could be included in the results sections or figure legend).” As suggested by Reviewer 3, we have included additional histological data in Supplementary Figure 2. In addition to the histology at 10-12 weeks of age, the new data includes histology of multiple skeletal muscle tissues (quadriceps, EDL, soleus) at one year of age. The histology of *Limch1 6exKO* tissue at different time points showed no morphological differences (centralized nuclei or fibrosis) consistent with no change in muscle weight which led us to emphasize the significant effect of mLIMCH1 knockout on skeletal muscle function in the absence of muscle loss or overt structural changes. In regard to fiber-type, we have included histology of both the EDL (fast-twitch) and soleus (slow-twitch) and even after one year, we observe no gross morphological differences. Additionally, we analyzed the force production of both the EDL and soleus (Figure 3) with the fiber-type predominance of these tissues in mind and found decreased force generation in both tissues. We included the types of skeletal muscle tissue analyzed and the age of the mice in Supplementary Figure 2 as per the reviewer’s suggestion.
- “The authors performed RNAseq analysis in the skeletal muscle of the KO mouse (Fig 2B). What is the result of this experiment? Is the KO muscle transcriptome different or similar to control muscles?” We conducted RNA-sequencing on tissue from HOM *Limch1 6exKO* and WT controls and the results were disappointing showing minor differences that did not contribute to understanding the phenotype. We used this data only to show the loss of the six exons in Fig. 2B, however, we decided that RT-PCR analysis was the better assay since it shows not only that the exons are not included but also that exons 8 and 15 are spliced correctly, which is not apparent using the RNA-seq displayed on the genome browser.

4. Description of analyses that authors prefer not to carry out

Please include a point-by-point response explaining why some of the requested data or additional analyses might not be necessary or cannot be provided within the scope of a

Revision Plan

revision. This can be due to time or resource limitations or in case of disagreement about the necessity of such additional data given the scope of the study. Please leave empty if not applicable.

Reviewer 1 (Both points listed as optional)

- “If the authors perform TEM, can they see defects in t-tubules or organization of the sarcoplasmic reticulum, that are not visible by light microscopy?” We considered conducting TEM to investigate sarcomeric, T-tubule, or sarcolemma changes in myofibers derived from HOM *Limch1* 6exKO mice, but we concluded that it would most likely be of limited use. We do not think that T-tubule structural changes will be observed via TEM primarily due to the challenges of finding significant changes compared to WT controls in which one can always find abnormal structures. In our experience and the experience of our collaborator (Dr. Rodney) the disruptions must be dramatic to distinguish from the noncanonical structures often observed. Thus, we do not plan on conducting TEM to identify defects in the T-tubules.
- “If the muscle-specific isoform is transfected or transduced into differentiated myotubes, how does this affect calcium dynamics in the culture system?” While an interesting idea, we do not plan on conducting this experiment for multiple reasons. One issue is that all of our data is derived from in vivo analysis or from isolated myofibers and our concern is that the relatively immature state of myotubes in culture will provide a poor comparison to isolated myofibers. Therefore, we believe that it will be difficult to add meaningful data to the calcium data presented in Figure 5 through this experiment. Additionally, we have observed mis-localization of the overexpressed uLIMCH1 and mLIMCH1 in C2C12 cells that we believe would add too many caveats for meaningful interpretation of the results, regardless of the effects on calcium dynamics

March 6, 2023

RE: Life Science Alliance Manuscript #LSA-2022-01868-TR

Dr. Thomas Cooper
Baylor College of Medicine
Pathology & Immunology
1 Baylor Plaza
Houston, Tx 77030

Dear Dr. Cooper,

Thank you for submitting your revised manuscript entitled "The role of Limch1 alternative splicing in skeletal muscle function". We would be happy to publish your paper in Life Science Alliance pending final revisions necessary to meet our formatting guidelines.

- please address Reviewer 2's remaining comment #3
- please add ORCID ID for secondary corresponding author-they should have received instructions on how to do so
- please add the Twitter handle of your host institute/organization as well as your own or/and one of the authors in our system
- please add a conflict of interest statement to the main manuscript text

To upload the final version of your manuscript, please log in to your account: <https://lsa.msubmit.net/cgi-bin/main.plex>. You will be guided to complete the submission of your revised manuscript and to fill in all necessary information. Please get in touch in case you do not know or remember your login name.

A. FINAL FILES:

B. MANUSCRIPT ORGANIZATION AND FORMATTING:

Sincerely,

Reviewer #1 (Comments to the Authors (Required)):

Summary

In their paper "The role of Limch1 alternative splicing in skeletal muscle function," Penna and colleagues report a muscle-specific isoform of Limch1 and investigate its function in skeletal muscle. They show that a muscle-specific isoform of Limch1 is expressed preferentially in mature muscle and localized to the sarcolemma, while the Limch1 isoform expressed in embryonic or immature muscle is predominantly cytoplasmic. Animals mutant for the muscle-specific isoform have reduced grip strength and force generation. Notably, although muscle structure and T-tubules are structurally not affected, mutant muscle shows evidence of disrupted calcium handling. Limch1 is also misspliced in DM1 and Mbnl1/2 double mutant mice, suggesting the muscle isoform is disease relevant and regulated by MBNL.

Comments

In this revised version of the manuscript, the authors have addressed all of my previous concerns. They included additional data in Figure 1, that makes the conservation and splicing patterns between human and mouse Limch1 clearer. They have revised the analysis and the text for Figure 4, making the localization of the two Limch1 isoforms clearer and more informative. The conclusions drawn from this portion of the results are now supported by the data. All minor comments on labeling and grammar have been addressed.

Significance

The revisions have clearly improved the manuscript. This is a well-written study identifying the function of a muscle-specific isoform of LIMCH1, as well as implicating a switch in Limch1 isoform expression in DM1 models as a target of MBNL regulation. It presents multiple new tools to study mLimch1, identifies a possible role for mLIMCH1 in calcium regulation, and is a definite advance in our understanding of developmentally-regulated splice isoform transitions that are disease relevant. The work will be of interest to the muscle, DM1 and splicing communities.

Reviewer #2 (Comments to the Authors (Required)):

This manuscript contributes new information concerning the role of alternative splicing in muscle development and function. The authors report an intriguing alternative splicing difference between fetal and adult tissues involving 6 (mouse) or 7 (human) consecutive exons in the LIMCH1 gene that are included predominantly in adult skeletal muscle to encode a longer isoform of the protein, mLIMCH1. These exons are excluded in the ubiquitous isoform uLIMCH1. CRISPR/Cas9-mediated deletion of these exons in mouse models was performed to explore functional consequences of loss of mLIMCH1 to muscle physiology. Muscles deficient in this isoform did not show overt differences in morphology or mass, but did exhibit changes in subcellular localization characterized by reduction of sarcolemma staining (of mLIMCH1) in wild type but increased cytoplasmic localization the remaining uLIMCH1 protein in knockouts. The knockout muscles suffered grip strength weakness in vivo, decreased force generation ex vivo, and defects in calcium handling. Mis-splicing of LIMCH1 in human patients with myotonic dystrophy type 1, and in mice with muscle knockout of MBNL, suggest that MBNL proteins are important regulators of LIMCH1 splicing.

COMMENTS

1. I reviewed this paper earlier for Review Commons, and find the current manuscript significantly improved. Previous issues have been addressed very well and I have only a few comments.
2. The major conclusions of the manuscript are clear and convincing -a muscle-specific cluster of exons in the LIMCH1 gene whose splicing is regulated directly or indirectly by MBNL splicing factor(s); loss of these exons compromises muscle strength; and these exons are poorly spliced in muscle of myotonic dystrophy patients. The data for these conclusions is strong.
3. The PCR results in Figure 1 comparing developmental splicing changes in mouse vs human are confusing: although both species are said to increase expression of the mLIMCH1 isoform in mature muscle, the PCR results show increases of the higher MW band in mature muscle in one case but decreases in the other. Also, is the mLIMCH1 isoform (containing the extra exons) really expected to yield a smaller PCR product than the uLIMCH1 isoform? I expect all can be clarified with additional information about the expected band sizes and by double-checking the labeling of the diagram.
4. Review of the paper was somewhat hampered because the figures weren't numbered and were presented in a different order than the corresponding figure legends (especially with regard to placement of the supplemental figures).

1. General Statements

We thank the reviewers from Review Commons for their thorough reviews of our manuscript entitled, “The role of *Limch1* alternative splicing in skeletal muscle function.” We were delighted by the many supportive comments of all three reviewers calling our study a “definite advance in our understanding of developmentally-regulated splice isoform transitions that are disease relevant”, “good comprehensive study with convincing results, the design of the experiments is good, and the conclusions are solid”, and “The article is well written, and I favor the publication of this [article] with minor revisions.”

The reviews include comments on the interest in identifying the mechanism of action of mLIMCH1 in skeletal muscle function such as “ [The study] presents multiple new tools to study mLimch1 and identifies a possible role for mLIMCH1 in calcium regulation, but stops short of identifying the mechanism by which this regulation occurs.” While we agree that how the skeletal muscle-specific isoform of LIMCH1 affects calcium handling is of interest, we respectfully suggest that this manuscript describe previously unknown biology that will be of interest to investigators in different fields including muscle physiology, alternative splicing regulation, and skeletal muscle pathology in myotonic dystrophy. All experiments in this manuscript are performed in vivo using skeletal muscle tissues from animals lacking the isoform of *Limch1* that is expressed only in skeletal muscle and is normally induced after birth. Comparisons were made to age-matched wild-type control animals, often litter mates. The results establish the functional significance of the LIMCH1 protein and particularly the muscle-specific isoform in skeletal muscle through extensive analysis of LIMCH1 localization and the impact of mLIMCH1 knockout on muscle strength, force generation, calcium handling and the disease relevance of this splicing transition in myotonic dystrophy type 1. Please review the comments of all three reviewers who were quite favorable to the significance of the work and overall favorable to its publication. Below, we clarify and describe additional data that has, and will be added to the manuscript to address all comments of the reviewers. Our responses are in blue font.

2. Description of the planned revisions

- All planned revisions have been included in the updated manuscript.

3. Description of the revisions that have already been incorporated in the transferred manuscript

Please insert a point-by-point reply describing the revisions that were already carried out and included in the transferred manuscript. If no revisions have been carried out yet, please leave this section empty.

Reviewer 1

- “Page 6 - data not shown. The point of conservation is not essential to this story, but the authors should either include a table or panel with that data, or remove the data not shown statement. Given the putative relevance to DM1, it might be

Revision Plan

preferable to include data to support the developmental transition in human data.” We agree that data showing the conserved LIMCH1 splicing transition in human skeletal muscle development will support this point. We included RT-PCR analysis of LIMCH1 splicing in fetal and adult human skeletal muscle RNA in Figure 1 to support the reversion of splicing to the fetal pattern observed in DM1. The results will complement the normal *Limch1* splicing transition in mice (Figure 1) and the normal and aberrant fetal splicing patterns shown for unaffected and DM1 adult skeletal muscle, respectively (Figure 6).

- “Figure 4 - The authors do a nice experiment to show the localization of *Limch1* and raise an antibody to detect the muscle specific isoform. The data seem to show that the muscle-specific isoform localizes to the sarcolemma, and this staining is largely lost in the mutant mice. **By contrast, one could infer that the cytoplasmic signal in the WT comes from the ubiquitous isoform (which accounts for 30-40% of the *Limch1* expression).** This is consistent with the validation in Fig. 2. However, the authors in the text claim this experiment reveals an increased distribution throughout the myofiber, or a more even signal distribution in the cytoplasm, and that the u*Limch1* cannot recapitulate m*Limch1* localization. Fig. 2 suggests that total levels of *Limch1* are increased (as noted by the authors in the discussion). **Given that the muscle-specific isoform localizes to the sarcolemma, and the ubiquitous isoform is presumably sarcoplasmic, it isn't clear to me that there is any change in localization per se.** What the authors show is just that the signal at the sarcolemma is lost, and if one compares the intensity in the right-hand plots in Fig. 4B, they are comparable in the sarcoplasmic region. It seems likely there is more of the ubiquitous isoform, and what is seen here is just how that isoform localizes. The quantification the authors perform in D would likely show this strong difference in the localization of the muscle isoform. If the authors redo this quantification, exclude the signal at the sarcolemma and normalize to the average pixel intensity in the fiber, do they still see a difference? I am not convinced that the "clustering" of the signal of the ubiquitous, cytoplasmic isoform is in any way changed. Given the difference in the two proteins, I also would not expect that the ubiquitous isoform could compensate for loss of the muscle isoform, and would not expect it to "recapitulate" the muscle-isoform localization.” We agree that Figure 4 and the explanation in the text was not clear and we thank the reviewer for pointing this out. We have addressed this concern by modifying the figure as suggested by the reviewer and clarifying the description in the results section. The main point, that is recognized by the reviewer but needed clarification, is that the mLIMCH1 isoform preferentially localizes to the sarcolemma and the uLIMCH1 isoform is preferentially cytoplasmic. In the HOM *Limch1* 6exKO myofibers, the increased cytoplasmic signal is due to the increased level of uLIMCH1 as shown by the western blot in Figure 2. The reviewer is correct that there is not a “change in localization of isoforms per se”. We clarified this point to highlight the differential localization of the uLIMCH1 and mLIMCH1 isoforms within the sarcolemma vs. the sarcoplasm. The revision of the plot profile in Figure 4B and the analysis of the standard deviation of signal in

Revision Plan

Figure 4D demonstrates the stark difference in staining observed between the HOM *Limch1* 6exKO and WT myofibers when stained with a pan-LIMCH1 antibody. The signal intensity plot profile from sarcolemma to sarcolemma (Figure 4B) indicates that the uLIMCH1 isoform is not “mis-localized” upon mLIMCH1 knockout as we originally (mis)-stated. Upon mLIMCH1 knockout, there is increased uLIMCH1 expression compared to WT myofibers. Considering this in combination with the sarcolemma preference of mLIMCH1 (Figure 4E) and the significant loss of signal in the sarcolemma region in *Limch1* 6exKO myofibers, we conclude that in HOM *Limch1* 6exKO myofibers, uLIMCH1 is primarily localized throughout the sarcoplasm.

Reviewer 1 (optional)

- “Experiments looking more closely at LIMCH1 co-localization with other proteins at the sarcolemma or the sufficiency of the muscle-specific region to localize would also be useful (for example, can the muscle-specific region localize GFP to the membrane in cells?).” We performed immunofluorescence microscopy of LIMCH1 with several skeletal muscle-relevant proteins but did not observe: (1) disruptions of normal structures in HOM *Limch1* 6exKO compared to WT myofibers or (2) colocalization that helped clarify any mechanistic role of mLIMCH1 or uLIMCH1. Therefore this data was not included in the original manuscript. In regard to the suggestion on the sufficiency of the muscle-specific region to localize to the sarcolemma region, we had previously generated a plasmid to express a fluorescent protein fused to the protein encoded by the six skeletal muscle-specific exons of LIMCH1 but it failed to localize to the sarcolemma. In collaboration with protein structural experts at Baylor College of Medicine, we analyzed the skeletal muscle-specific region of LIMCH1 and found it to be entirely disordered without known homologs. It appears that this region has no secondary structure but when expressed within the entire LIMCH1 protein which has conserved domains (calponin homology, LIM, coiled-coil regions) and upon protein binding, it is possible for the region to adopt a structure facilitating its binding in the sarcolemma region. Therefore we believe that regions common to both isoforms are required in combination with the muscle-specific region for preferential localization to the sarcolemma.

Reviewer 1 (minor comments)

- “In the Figure 3 legend, the order of the descriptions for B-C and D-E is switched. The order of the panels matches the text, but the legend switches the description of the force-frequency curves (shown in B & C but labeled as D & E), with the description of the rate of relaxation and contraction plots (shown in D and E but labeled as B and C in the legend).” We fixed this error and thank the reviewer for pointing it out.
- “The scale in Figure 4, panel B between the top and bottom plots is not the same, so it is difficult to compare, particularly for the panels on the right. See comment above.” In addition to clarifying uLIMCH1’s localization upon mLIMCH1 knockout within the text, we added figure titles above the plot profile which will

Revision Plan

clarify the different plot profiles for the reader. In regard to the comment about the scale of the plot profile, we have addressed this by re-scaling the two plot profiles on the right in Figure 4B. These plot profiles now share the same scale, which is advantageous because this plot profile better emphasizes the stark difference in signal observed between the sarcolemma and sarcoplasm in WT myofibers that is lacking in HOM *Limch1* 6exKO myofibers.

Reviewer 2

- “Figure 6A: There is a discrepancy between gene structures and splicing isoforms shown in Fig. 1 vs Fig. 6. There are differences in spacing between exons, and there appear to be six exons in the differentially regulated region in Fig 1, but seven exons in Fig 6. Perhaps this is a difference between human and mouse genes? Does the human gene actually regulate seven exons in this region, rather than six exons in the mouse? In both figures the gene is labeled as *Limchi1*, and both figures indicate that the ubiquitous isoform lacks exons 9-14. Please clarify.” The reviewer is correct that the human mLIMCH1 isoform contains seven exons that are skeletal muscle-specific compared with the six exons that are skeletal muscle-specific in the mouse. The seven human exons encode 544 amino acids with 65% homology with the mouse segment. We have clarified this in the figure legend and text. Exons 9-14 are shown in Figure 6B since this diagrams the mouse gene.
- “The methods section on RT-qPCR and RNA splicing presumably refers to analysis of mouse tissues. What is the origin of the human DM1 RNA-seq data?” We obtained adult human DM1-affected and non-affected skeletal muscle autopsy samples from colleagues and the NDRI and performed RNA-sequencing at Baylor College of Medicine. The RNA-seq has not yet been published, but we include the data for *LIMCH1* to demonstrate the dramatic change in the alternative splicing pattern in DM1 skeletal muscle tissue. This has been clarified in the methods section.
- “Perhaps the word “activity” should be deleted in the following sentence: “The sole study investigating the function of LIMCH1 characterized it as an actin stress fiber associated protein that binds non-muscle myosin 2A (NM2A) activity to regulate focal adhesion formation.” We thank the reviewer for pointing this out and we have removed this word.
- “The PCR results in Figure 1 comparing developmental splicing changes in mouse vs human are confusing: although both species are said to increase expression of the mLIMCH1 isoform in mature muscle, the PCR results show increases of the higher MW band in mature muscle in one case but decreases in the other. Also, is the mLIMCH1 isoform (containing the extra exons) really expected to yield a smaller PCR product than the uLIMCH1 isoform? I expect all can be clarified with additional information about the expected band sizes and by double-checking the labeling of the diagram.” We thank the reviewer for pointing out this mistake. The horizontal label of the human RT-PCR in Figure 1E was

Revision Plan

swapped. This has been fixed and now correctly shows the developmental alternative splicing pattern of human *LIMCH1*. In regard to the size of the PCR products, the products are obtained by a combination of three different primers where the forward primer F1, and the reverse primer R1, provide the *mLimch1* product. While primers F1 and R2 provide the *uLimch1* product. Because of the large size of the region between the binding sites of F1 and R2, and since the polymerase we are using is not suitable for long-range PCR, this *uLimch1* product will only be obtained in the absence of the skeletal muscle specific region. Therefore, the product sizes will be dependent upon the primer design which differs for mouse and human RNA with the expected product sizes now included in the methods section.

Reviewer 3

- “The diminution of the muscle force production in *Limch16exKO* is not correlated with a change in morphology of the myofibers in H&E and picosirius stainings (Fig S2). Did the authors look at other skeletal muscles, fiber type, size, or different time points? (The age of the mouse and the name of the skeletal muscle used for the histology could be included in the results sections or figure legend).”
As suggested by Reviewer 3, we have included additional histological data in Supplementary Figure 2. In addition to the histology at 10-12 weeks of age, the new data includes histology of multiple skeletal muscle tissues (quadriceps, EDL, soleus) at one year of age. The histology of *Limch1 6exKO* tissue at different time points showed no morphological differences (centralized nuclei or fibrosis) consistent with no change in muscle weight which led us to emphasize the significant effect of *mLIMCH1* knockout on skeletal muscle function in the absence of muscle loss or overt structural changes. In regard to fiber-type, we have included histology of both the EDL (fast-twitch) and soleus (slow-twitch) and even after one year, we observe no gross morphological differences. Additionally, we analyzed the force production of both the EDL and soleus (Figure 3) with the fiber-type predominance of these tissues in mind and found decreased force generation in both tissues. We included the types of skeletal muscle tissue analyzed and the age of the mice in Supplementary Figure 2 as per the reviewer’s suggestion.
- “The authors performed RNAseq analysis in the skeletal muscle of the KO mouse (Fig 2B). What is the result of this experiment? Is the KO muscle transcriptome different or similar to control muscles?” We conducted RNA-sequencing on tissue from HOM *Limch1 6exKO* and WT controls and the results were disappointing showing minor differences that did not contribute to understanding the phenotype. We used this data only to show the loss of the six exons in Fig. 2B, however, we decided that RT-PCR analysis was the better assay since it shows not only that the exons are not included but also that exons 8 and 15 are spliced correctly, which is not apparent using the RNA-seq displayed on the genome browser.

Revision Plan

4. Description of analyses that authors prefer not to carry out

Please include a point-by-point response explaining why some of the requested data or additional analyses might not be necessary or cannot be provided within the scope of a revision. This can be due to time or resource limitations or in case of disagreement about the necessity of such additional data given the scope of the study. Please leave empty if not applicable.

Reviewer 1 (Both points listed as optional)

- “If the authors perform TEM, can they see defects in t-tubules or organization of the sarcoplasmic reticulum, that are not visible by light microscopy?” We considered conducting TEM to investigate sarcomeric, T-tubule, or sarcolemma changes in myofibers derived from HOM *Limch1* 6exKO mice, but we concluded that it would most likely be of limited use. We do not think that T-tubule structural changes will be observed via TEM primarily due to the challenges of finding significant changes compared to WT controls in which one can always find abnormal structures. In our experience and the experience of our collaborator (Dr. Rodney) the disruptions must be dramatic to distinguish from the noncanonical structures often observed. Thus, we do not plan on conducting TEM to identify defects in the T-tubules.
- “If the muscle-specific isoform is transfected or transduced into differentiated myotubes, how does this affect calcium dynamics in the culture system?” While an interesting idea, we do not plan on conducting this experiment for multiple reasons. One issue is that all of our data is derived from in vivo analysis or from isolated myofibers and our concern is that the relatively immature state of myotubes in culture will provide a poor comparison to isolated myofibers. Therefore, we believe that it will be difficult to add meaningful data to the calcium data presented in Figure 5 through this experiment. Additionally, we have observed mis-localization of the overexpressed uLIMCH1 and mLIMCH1 in C2C12 cells that we believe would add too many caveats for meaningful interpretation of the results, regardless of the effects on calcium dynamics

March 14, 2023

RE: Life Science Alliance Manuscript #LSA-2022-01868-TRR

Dr. Thomas Cooper
Baylor College of Medicine
Pathology & Immunology
1 Baylor Plaza
Houston, Tx 77030

Dear Dr. Cooper,

Thank you for submitting your Research Article entitled "The role of Limch1 alternative splicing in skeletal muscle function". It is a pleasure to let you know that your manuscript is now accepted for publication in Life Science Alliance. Congratulations on this interesting work.

DISTRIBUTION OF MATERIALS:

Again, congratulations on a very nice paper. I hope you found the review process to be constructive and are pleased with how the manuscript was handled editorially. We look forward to future exciting submissions from your lab.

Sincerely,
